



# Wind and Topography Underlie Correlation Between Seasonal Snowpack, Mountain Glaciers, and Late-Summer Streamflow

Elijah N. Boardman[1,2], Andrew G. Fountain[3], Joseph W. Boardman[4,5], Thomas H. Painter[5], Evan W. Burgess[5], Laura Wilson[6], Adrian A. Harpold[1,7]

[1]Graduate Program of Hydrologic Studies, University of Nevada, Reno, Reno, Nevada, 89557, USA
[2]Mountain Hydrology LLC, Reno, Nevada, 89503, USA
[3]Department of Geology, Portland State University, Portland, Oregon, 97207, USA
[4]Analytical Imaging and Geophysics LLC, Boulder, Colorado, 80305, USA
[5]Airborne Snow Observatories, Inc., Mammoth Lakes, California, 93546, USA
[6]Department of Earth Sciences, Dartmouth College, Hanover, New Hampshire, 03755, USA
[7]Department of Natural Resources and Environmental Science, University of Nevada, Reno, Reno, Nevada, 89557, USA

*Correspondence to*: Elijah N. Boardman (eli.boardman@mountainhydrology.com)

**Abstract.** In a warming climate, net mass loss from perennial snow and ice (PSI) contributes a temporary source of unsustainable streamflow. However, the role of topography and wind in mediating the streamflow patterns of deglaciating

watersheds is unknown. We conduct lidar surveys of seasonal snow and PSI elevation change for five adjacent watersheds in the Wind River Range, Wyoming (WRR). Between 2019 and 2023, net mass loss from PSI is equivalent to ~10-36% of August-September streamflow. Across 338 manually classified PSI features >0.01 km$^2$, glaciers contribute 68% of the total mass loss, perennial snowfields contribute 8%, rock glaciers contribute 1%, buried ice contributes 6%, and the remaining 17% derives from semi-annual snowfields and small snow patches. Surprisingly, watersheds with more area-normalized

seasonal snow produce less late-summer streamflow (r = -0.60), but this correlation is positive (r = 0.88) considering only deep snow storage (SWE >2 m). Most deep snow (87%) is associated with favorable topography for wind drift formation. Deep seasonal snow limits the mass loss contribution of PSI features in topographic refugia. We show that watersheds with favorable topography exhibit deeper seasonal snow, more abundant PSI features (and hence greater mass loss during deglaciation), and elevated late-summer streamflow. As a result of deep seasonal snow patterns, watersheds with the most

abundant PSI would still produce 45-78% more late-summer streamflow than nearby watersheds in a counterfactual scenario with zero net mass loss. Similar interrelationships may be applicable to mountain environments globally.

## 1 Introduction

Climate warming reduces snow and ice storage in mountain environments, disrupting historical water supply timing and affecting downstream ecosystems (Milner et al., 2017; Drenkhan et al., 2023). Seasonal snow storage (i.e., snow that

accumulates and melts each year) provides a seasonal buffer between the timing of precipitation and runoff (Barnett et al., 2005; Stewart et al., 2005; Li et al., 2017; Gordon et al., 2022). Perennial snow and ice (PSI) features, including glaciers,





perennial snowfields, rock glaciers, and buried ice, collectively provide a decadal to century scale buffer that reduces interannual streamflow variability between wet and dry years (Meier, 1969; Fountain and Tangborn, 1985; Moore, 1992; Pohl et al., 2017; van Tiel et al., 2020). However, climate warming is causing earlier seasonal snowmelt and driving glacier

recession globally, which reduces warm-season water availability and increases temporal variability (Oerlemans, 1986; Zemp et al., 2009; Huss, 2012; Zemp et al., 2015; Brun et al., 2017; Huss et al., 2017; Shean et al., 2020; Hugonnet et al., 2021). Earlier snowmelt and glacier recession both contribute to the snow-albedo feedback by reducing landscape albedo, which increases the energy available for melt and further accelerates the reduction of snow and glacier storage (Johnson and Rupper, 2020; Di Mauro and Fugazza, 2022; Davies et al., 2024). This feedback process is a driver of elevation dependent

warming (McGuire et al., 2012; Rangwala and Miller, 2012; Pepin et al., 2015; Palazzi et al., 2019). Glacier recession poses challenges for socio-hydrologic adaptation in downstream communities (Kaser et al., 2010; Carey et al., 2017; Drenkhan et al., 2023). With reduced seasonal snow and fewer PSI features to buffer precipitation variability, mountain watersheds are increasingly vulnerable to hydrological extremes including snow drought (Harpold et al., 2017; Dierauer et al., 2019; Huning and AghaKouchak, 2020).

Rough mountain topography can sustain PSI features at or below the regional equilibrium line altitude (ELA) due to wind drifting, avalanches, and topographic shading, making it difficult to generalize mass balance patterns across a mountain landscape (Kuhn, 1995; Allen, 1998; Hoffman et al., 2007; Hughes 2010; Florentine et al., 2018; 2020). Estimates of mountain PSI mass loss at the landscape scale are generally based on remote sensing (e.g., Bamber and Rivera, 2007) or

extrapolation (e.g., Huss, 2012). Optical remote sensing is useful for mapping spatial patterns of seasonal snow persistence and PSI extent (Meier, 1980; Selkowitz and Forster, 2016; Hammond et al., 2018), but these techniques do not provide a direct estimate of mass or volume changes. Repeat lidar remote sensing can quantify elevation changes caused by snow accumulation and PSI ablation, constraining the spatial variability and timing of PSI mass loss (Sold et al., 2013; Helfricht et al., 2014).


Net mass loss from PSI features can contribute a considerable fraction of streamflow in alpine regions (Schaner et al., 2012; Bliss et al., 2014; Huss and Hock, 2018). PSI mass loss temporarily increases mountain streamflow by providing an additional source of late-summer meltwater (Lambrecht and Mayer, 2009; Huss and Hock, 2018). However, this increased streamflow is unsustainable because PSI features are finite, ultimately leading to streamflow reductions (Stahl and Moore,

2006; Nolin et al., 2010; Moore et al., 2020; Rets et al., 2020). The fractional mass loss contribution to annual streamflow depends on the relative volume of meltwater contributed by net mass loss versus annual precipitation (O'Neel et al., 2014).

The impact of deglaciation on mountain streamflow patterns is confounded by spatial patterns of seasonal snow persistence. Seasonal snow persistence is a primary driver of hydrograph shape and streamflow volume in snowy alpine watersheds (Hall

et al., 2012; Richer et al., 2013; Barnhart et al., 2016; Hammond et al., 2018). Hydrograph shape determines the timing of





water availability, which is of paramount importance in regions with limited artificial storage capacity (McNeely et al., 2018; Siirila-Woodburn et al., 2021; Gordon et al., 2024). The presence of PSI results from historic snow accumulation exceeding melt, so PSI features necessarily coincide with the most persistent areas of the seasonal snowpack. PSI features also mediate snowmelt runoff timing (Fountain and Tangborn, 1985; Fountain, 1996; Fleming and Clarke, 2005), so it can be challenging

to attribute seasonal hydrograph shape and inter-watershed variability exclusively to differences in seasonal snowmelt or PSI mass loss (Frenierre and Mark, 2014; Brahney et al., 2017). Paired watershed studies with variable PSI abundance can provide compelling evidence for the role of PSI features in shaping the downstream hydrograph (Hood and Berner, 2009; Bell et al., 2012; Kneib et al., 2020; Zhou et al., 2021). However, the causal relationship between snow persistence and PSI abundance complicates the interpretation of space-for-time studies of deglaciation. The persistence of cold-water stream

species after deglaciation supports the role of seasonal snow in sustaining alpine streamflow patterns (Muhlfeld et al., 2020).

The interaction of wind and mountain topography is a critical control on snow accumulation, which mediates both snowmelt runoff and the PSI surface mass balance (Dadic et al., 2010; Clark et al., 2011; Freudiger et al., 2017). Processes involved in snow-wind interactions include orographic precipitation along downwind storm tracks (e.g., Dettinger et al., 2004),

preferential deposition of snowfall in wind-sheltered areas (e.g., Lehning et al., 2008), sublimation of blowing snow (e.g., Strasser et al., 2008; but see also Groot Zwaaftink et al., 2013), and snow drift formation from scour and redistribution (e.g., Elder et al., 1991). Wind-induced snow depth heterogeneity can delay snowmelt runoff (Luce et al., 1998; Hartman et al., 1999; Marsh et al. 2024) because the smaller surface area of deep snow drifts reduces incoming radiation and turbulent heat fluxes compared to an equal amount of snow spread thinly (Schneider et al., 2020). The combination of snow accumulation

through drifting and avalanches can create areas of deep snow that can survive multiple summers, sustaining small PSI features in otherwise non-glacial environments (Huss and Fischer, 2016; Mott et al., 2019). Glaciers close to the regional ELA have a tendency to face east of poleward, consistent with snow drift formation related to global wind directions, especially in areas with large, smooth upwind contributing areas (Evans, 1977; 2006).

To explore how wind and topography mediate the interrelationship of snow, PSI mass loss, and streamflow patterns, we leverage repeat airborne lidar surveys across the Wind River Range, Wyoming, (WRR), an alpine mountain range that exhibits a variety of PSI features and topographic refugia. By analyzing spatially extensive snow and geodetic mass balance data across five adjacent watersheds, we address the following research questions: (1) What fraction of late-summer streamflow derives from PSI mass loss in the WRR? (2) How do wind and topography mediate spatial patterns of snow

accumulation and PSI mass loss? (3) Are inter-watershed streamflow patterns primarily the result of transient deglaciation or topographically mediated seasonal snowmelt?





## 2 Regional Context

Adjacent watersheds in the Wind River Range, Wyoming, (WRR) exhibit differing streamflow patterns, variable snow persistence, and variable PSI abundance within a compact geographic area, providing a natural experiment on the interrelationship of these factors (Hall et al., 2012; Bell et al., 2012). The WRR includes the largest glaciers in the U.S. Rocky Mountain region, with a total glacier and perennial snowfield area (surficial features >0.01 km$^2$ only) around 26 km$^2$ in 2015 (Fountain et al., 2023). Runoff from these glaciers contributes to the headwaters of the Wind River in the Missouri River Basin and the Green River in the Colorado River Basin. WRR streamflow is an important water source for regional communities and ecosystems (Cheesbrough et al., 2009; MacKinnon, 2015). The WRR watershed with the greatest current PSI abundance, Dinwoody Creek, historically provides a reliable source of late-summer water that is more robust to local snow drought compared to nearby watersheds with fewer PSI features (McNeely et al., 2018).

Glaciological research in the WRR began early in the past century (Wentworth and Delo, 1931), with many subsequent studies mapping glacier areas (e.g., Meier, 1951; Thompson et al., 2011; Maloof et al., 2014; DeVisser and Fountain, 2015; Marks et al., 2015) and several studies measuring thinning rates (e.g., Marston et al., 1991; Naftz and Smith, 1993; VanLooy et al., 2013). The WRR glaciers have been generally in retreat since the Little Ice Age maximum around 1900 (Meier, 1951), with about a 44% reduction in area since that time (DeVisser and Fountain, 2015). Climate warming is contributing to PSI mass loss, with alpine areas of the WRR warming by 3.5 °C between the mid-1960s and early 1990s based on ice core measurements (Naftz et al., 2002). Based on aerial photogrammetry and ice-penetrating radar, Marston et al. (1991) suggested that one of the WRR's largest glaciers (Dinwoody Glacier) could vanish by 2016 if mass loss continued at the 1958-1983 rate. However, the Dinwoody Glacier only decreased in area by 12% between 1989 and 2009 (DeVisser and Fountain, 2015), highlighting the uncertainty and potential nonstationarity of mountain glacier recession rates. More recent ice-penetrating radar measurements of the Continental Glacier by VanLooy et al. (2014) suggest that at least some of its ice mass will likely persist several hundred years if the 1966-2012 thinning rate remains stationary.

PSI mass loss contributes to WRR streamflow, but different approaches to quantifying this contribution on different time periods provide variable results (e.g., Marston et al., 1989; DeVisser and Fountain, 2015). Watersheds with greater PSI abundance exhibit elevated late-summer streamflow, but not all of the difference in streamflow generation among watersheds is a result of net mass loss (Bell et al., 2012). Studies based on area-volume scaling relationships tend to estimate that PSI mass loss contributes roughly 1-10% of the August-September streamflow volume, and no higher than 23% (Cheesbrough et al., 2009; Thompson et al., 2011; Maloof et al., 2014; DeVisser and Fountain, 2015; Marks et al., 2015). Studies based on stable isotope unmixing tend to provide higher estimates, e.g., 53-70% of August streamflow in the most-glaciated watershed and 37-83% of August streamflow for two less-glaciated watersheds (Cable et al., 2011; Vandeberg and VanLooy, 2016; VanLooy and Vandeberg, 2019; 2024). Measurements of surface runoff from glaciers indicate a 14-33%




contribution to August streamflow in two watersheds (Vandeberg and VanLooy, 2016; VanLooy and Vandeberg, 2019). Reducing uncertainty in the PSI mass loss contribution to streamflow is important to constrain the climate vulnerability of the mountain water supply. Moreover, most estimates of PSI mass loss are too small to explain observed differences in late-summer streamflow among WRR watersheds with different PSI area fractions (Bell et al., 2012), provoking a question of what underlying processes might drive inter-watershed differences in both PSI abundance and late-summer streamflow.


Inter-watershed patterns of glaciation in the WRR are consistent with expected patterns of deep snow accumulation inferred from avalanches and wind transport across topographic divides (Baker, 1946). The geomorphology of the WRR is characterized by so-called "biscuit-board" topography (Hobbs, 1912), with cirques incised into high-elevation plateaus (Westgate and Branson, 1913; Blackwelder, 1915; Mears, 1993; Anderson, 2002). On these high-elevation, relatively

smooth erosional surfaces, snowfall is transported downwind, accumulating in cirque basins (Graf, 1976) and sustaining PSI features at elevations below the regional ELA, similar to "drift glaciers" observed in the Colorado Front Range (Outcalt and MacPhail, 1965; Meierding, 1982; Hoffman et al., 2007). Due to topographic shading and shelter from the prevailing wind, north-facing and east-facing aspects support more abundant and larger PSI features (Fig. 7 of DeVisser and Fountain, 2015), contributing to greater PSI abundance in watersheds on the east (downwind) side of the WRR.

**3 Methods**

We estimate snow and ice mass changes in five adjacent WRR watersheds with repeat airborne lidar surveys. Using the geodetic method (e.g., Fischer, 2011), we map PSI mass changes by comparing lidar elevation models acquired in 2019 and 2023 with density assumptions based on literature and regional field measurements. We also use lidar to map seasonal snow accumulation near peak SWE in 2024 with snow density constrained by a statistical model based on regional field

measurements. Streamflow is measured at five gauges (Fig. 1, Table 1). The watershed-average statistics in Table 1 reflect the arbitrary location of each stream gauge relative to its headwaters, since gauges that are farther downstream have much larger watershed areas without meaningfully increased streamflow. We control for this effect in subsequent analysis of watershed-scale snow metrics by only considering the area above 3000 m, which approximates the snow line at the time of the 2024 lidar survey.






**Figure 1: Relief map of the northern Wind River Range, showing perennial snow and ice features considered in this study, watershed boundaries, and stream gauge locations.**





| Watershed | USGS Stream Gauge # | Area (km²) | Area Above 3000 m (km²) | PSI Area Fraction | Elevation Median and 90th Percentile (m) | Temperature DJF, MAM, JJA, SON (°C) | Precip. (mm yr⁻¹) | Streamflow (mm yr⁻¹) (Q / P) |
|---|---|---|---|---|---|---|---|---|
| Dinwoody | 06221400 | 228 | 166 | 7.3% | 3260, 3730 | -9, -2, 10, 0 | 689 | 571 / 0.83 |
| Torrey | N/A | 125 | 88 | 5.3% | 3210, 3680 | -9, -2, 9, 0 | 727 | 445 / 0.61 |
| Bull Lake | 06224000 | 484 | 333 | 3.5% | 3170, 3580 | -8, -1, 10, 1 | 694 | 517 / 0.75 |
| Pine | 09196500 | 196 | 154 | 1.8% | 3200, 3510 | -8, -1, 10, 0 | 877 | 768 / 0.88 |
| Upper Green | 09188500 | 1212 | 352 | 0.9% | 2750, 3420 | -9, 0, 11, 1 | 649 | 346 / 0.53 |

**Table 1: Geographic and climatological metrics for the five gauged study watersheds, arranged in order of decreasing perennial snow and ice (PSI) area fraction. DJF = December January February, MAM = March April May, JJA = June July August, SON = September October November. Q / P = streamflow / precipitation. Climate data are the gridMET 1991-2020 mean (Abatzoglou, 2013). Streamflow is the 2005-2024 mean (U.S. Geological Survey, 2024a).**

### 3.1 Lidar Surveys

The U.S. Geological Survey (USGS) 3DEP program acquired lidar data at 1 m grid resolution over the northern Wind River Range PSI study area on August 17 and 18, 2019 (U.S. Geological Survey, 2019). Airborne Snow Observatories (ASO) acquired lidar data at 3 m grid resolution over the same area on October 7, 2023, which also includes color aerial photography at equal resolution (Painter et al., 2016). The difference in snow and ice storage between these two lidar acquisitions defines our measure of PSI net change, and areas with elevation changes between the lidar acquisitions define our concept of the active PSI area throughout this study.

Lidar datasets are reconciled and differenced to estimate elevation changes. We reproject and bilinearly resample the USGS 1 m digital terrain model (DTM) to match the ASO DTM. Coregistration is necessary to minimize errors from grid mismatch. We follow the approach of Nuth and Kääb (2011), considering both horizontal shifts and an elevation-dependent bias. Our application takes advantage of parallel computing to simultaneously solve for the horizontal shift and elevation-dependent bias using numerical optimization (optimParallel library: Gerber and Furrer, 2019). To define the control areas for coregistration, we delineate 51 landscape areas (10.6 km² total) with minimal snow or vegetation cover based on the ASO aerial photography. The coregistration areas span an elevation range of 2760 to 4120 m (median 3170 m), slope range of 0° to 81° (median 22°), and all aspects. The numerical optimization reduces the mean absolute vertical error to 14 cm between the two DTMs, with a median elevation difference of 0.0 cm and an interquartile range of -6.3 cm to +8.9 cm. Steeper slopes



tend to have higher errors (Pearson correlation r = 0.39 between slope and absolute error), and the mean absolute error increases to 24 cm for slopes >30° (31% of coregistration area).

We calculate the 2019-2023 DTM difference over the full survey area, then use a multi-step approach to identify changes that are likely associated with snow and ice. The USGS lidar data are classified by the lidar vendor using proprietary
software based partially on lidar return intensity (Woolpert, 2020). Using this classification, we include elevation changes within all areas classified as snow-covered in the August 2019 data with the exception of frozen lakes, which are manually excluded. Bare glacier ice, rock glaciers, and buried ice features are not adequately captured by the lidar snow classification, requiring manual delineation of additional areas that exhibit spatially coherent patterns of elevation change, as demonstrated by Robson et al. (2022). We find that a ±25 cm elevation change threshold helps refine the edges of manually delineated
polygons to more closely match the boundaries of apparent PSI features. All PSI features identified from elevation changes are compared with aerial imagery from Google Earth and the 2023 ASO acquisition to evaluate the likelihood of PSI occurrence in that location. In particular, we are careful to exclude steep rock features such as cliffs and sharp ridges, which cause spurious elevation differences due to lidar sampling uncertainty and slight grid mismatch.

We measure seasonal snow depth at 3 m resolution using lidar data collected by ASO on May 31, 2024. Snow depth is calculated relative to the updated October 7, 2023 PSI surfaces, and the identification of snow-covered areas is aided by imaging spectroscopy (Painter et al., 2016). Regional field measurements of snow temperature around the time of the lidar survey show a snowpack ripening elevation (0° snow temperature) around 3300 m, below 99% of the PSI area. The following week, we observed an increase in air temperatures with no additional snow accumulation. Thus, the 2024 snow
lidar survey represents approximately peak SWE conditions at PSI elevations.

### 3.2 Density Constraints

Although relative variations in snowpack density are typically an order of magnitude smaller than variations in snow depth, it is still important to constrain density variations across the landscape (Alford, 1967; Bormann et al., 2013; Wetlaufer et al., 2016), especially in deep drifts, where wind slab formation and metamorphism increase density (Tabler, 1980; Colbeck,
1982; de Leeuw et al., 2023). We constrain snow density variations with field data from nearby regions of the WRR spanning gradients of elevation, topography, and forest canopy cover. Vertically integrated bulk snow density data are available from 20 snow pit profiles across the six days before the lidar survey to one day afterward (May 25 to June 1, 2024). We also consider eight snow pit profiles from the prior year at the same point in the season (May 30 and 31, 2023). Locations with repeat measurements in 2023 and 2024 show consistent densities (0-7% difference) where the snow is
isothermal. Thus, we include five of the 2023 measurements in our 2024 density model since those pit locations are below the snowpack ripening elevation in both years. The deepest snow pit (from 2023) extends to 485 cm depth without reaching the ground; the density profile for this pit is approximately uniform with depth (0.57 g cm$^{-3}$ mean), so we extrapolate the



mean density to the full depth of ~580 cm indicated by lidar. Of the 25 total snow pits used to constrain density, 14 are above treeline and the other 11 span a range of forest canopy cover from near 0% to 99% (RCMAP data: Rigge et al., 2021). We additionally include 14 snow density measurements from five SNOTEL sites during the same survey period, excluding days with apparent snow bridging or similar anomalies (Natural Resource Conservation Service, 2024).

The highest snow densities are measured in deep wind drifts, with relatively lower densities at higher elevations and lower densities under forest canopies. We did not detect an aspect dependence for density during our May-June snow surveys. We test a variety of empirical model structures, ultimately finding that a simple parametric equation is best suited to extrapolate snow density in the study area (Appendix A Eq. A1-A4). Machine learning approaches such as Random Forest or Gaussian Process regression introduce unphysical irregularities in the density field as a result of over-fitting our small sample size (N = 39). Additionally, a parametric model structure allows us to include process knowledge, such as the snowpack ripening elevation and monotonically increasing bulk density in areas with greater snow depth (when all other variables are equal). Residuals from partial regression plots suggest a linear relationship between snow density and depth, a sigmoid relationship with elevation as a result of the snowpack ripening elevation, and an inverted hyperbolic relationship with forest canopy cover (small changes in canopy cover are most important close to zero).

We use Bayesian sampling of our empirical model to extrapolate measured snow densities across the landscape (Appendix A Eq. A5). We sample likely model parameters using Hamiltonian Monte Carlo implemented in Stan (Stan Development Team, 2023) with error weights based on an importance score for each density measurement. The importance score is calculated by determining how often each density measurement is the closest match to the normalized predictors (depth, elevation, and canopy cover) of $10^6$ grid cells randomly sampled from the lidar survey domain with a probability equal to the grid cell snow depth. This weighted model fitting strategy has the effect of reducing uncertainty in total SWE by preferentially fitting the model to density measurements that are representative of spatially extensive and/or locally deep areas of the snowpack. Across all 39 snow density measurements, the model achieves a root mean square error (RMSE) of 0.031 g cm⁻³, which is 7% error relative to the 0.43 g cm⁻³ mean across measurements (Supplemental Fig. S3). Considering that 25 of the 39 measurements are from forested areas with similar bulk densities and a low total sum of squares (low variance), the model $R^2$ for all 39 measurements is only 0.58. However, considering just the 14 measurements above treeline or just the 12 measurements with at least 1 m of snow depth, the model $R^2$ increases to 0.93 or 0.85, respectively. The fitted model indicates a maximum rate of density change with respect to elevation at 3320 m, consistent with regional observations of the snowpack ripening elevation (~3300 m based on snow temperature). We map snow density using the mean of 100 Bayesian samples for each 3 m grid cell, clipped to a range of 0.30-0.60 g cm⁻³ (Supplemental Fig. S4).

We assume a range of constant densities to convert the lidar-based 2019-2023 PSI volume change into mass (Table 2) based on our classification of PSI features (Sect. 3.3). For glaciers, rock glaciers, and buried ice, we adopt the value of 0.85 g cm⁻³





with an uncertainty range of ±0.06 g cm⁻³ (Sapiano et al., 1998; Huss, 2013). Although the volume of rock glaciers includes ice and rock, the rock volume is conserved, so we assume that elevation changes represent an equivalent depth of ice ablation. We assume that snowfields have a lower density than glaciers. One late-summer snow density measurement from

August 20, 2022, indicates a bulk density of 0.58 g cm⁻³ for a semi-annual snowfield at 3480 m elevation. Based on this measurement and the range of literature values for shallow firn (e.g., Sharp, 1951; Ambach and Eisner, 1966; Stevens et al., 2024), we assume a density of 0.60 g cm⁻³ for semi-annual snowfields and 0.70 g cm⁻³ for perennial snowfields, both with an uncertainty of ±0.10 g cm⁻³. The assumption of a uniform density within each glacier or snowfield may impact spatial patterns of mass loss at the 3 m grid scale, but we expect that comparisons among PSI features and watersheds should be

more robust to this assumption due to the spatial averaging of accumulation and ablation-zone density variations (Sapiano et al., 1998).

| Classification | Mean Density (g cm⁻³) | Density Range (g cm⁻³) | Basis of Estimate |
|---|---|---|---|
| Glaciers | 0.85 | 0.79-0.91 | Sapiano et al., 1998; Huss, 2013 |
| Rock Glaciers and Buried Ice | 0.85 | 0.79-0.91 | Assumed same as glaciers |
| Perennial Snowfields | 0.70 | 0.60-0.80 | Photos show mixed ice and snow/firn content; assumed intermediate density between glaciers and semi-annual snowfields |
| Semi-Annual Snowfields | 0.60 | 0.50-0.70 | Regional measurements and firn densification literature (e.g., Sharp, 1951; Ambach and Eisner, 1966; Stevens et al., 2024) |
| Snow Patches (<0.01 km²) | 0.60 | 0.50-0.70 | Assumed same as semi-annual snowfields |

**Table 2: Density assumptions for converting geodetic volume measurements into perennial snow and ice mass loss estimates.**

**3.3 Mass Loss and SWE Metrics**

Net mass loss from PSI is assumed to occur mostly in the late summer in the WRR (e.g., DeVisser and Fountain, 2015).

Satellite imagery and field experience both suggest that seasonal snow usually covers most of the PSI area through July. Mean daily air temperatures (Abatzoglou, 2013) at PSI elevations drop below freezing between September (mean 3.8 °C) and October (mean -2.3 °C), which ends the ablation season. Although the August 17-18, 2019, and October 7, 2023, lidar surveys are separated by ~4.1 years, we assume that the difference between these two surveys represents 4.5 years of ablation because the 2019 data are about halfway through the presumed ablation season and the 2023 data represent late-

season conditions. Variations in the persistence of the seasonal snowpack and differences in the timing of lidar surveys both





contribute uncertainty to our estimates of the PSI mass loss rate, but we constrain this uncertainty based on the 2024 snow data (Sect. 2.4).

To estimate the potential impact of deglaciation on streamflow, we define the "mass loss contribution" as the fraction of total annual meltwater derived from net mass loss. A PSI feature with a zero or positive mass balance would have a mass loss contribution of zero, since its meltwater would be entirely derived from seasonal snow and sustainable ice flow. We estimate the mass loss contribution for each PSI feature by dividing its annual mass loss by the sum of mass loss and seasonal snow mass. We emphasize that this metric is primarily intended for inter-feature comparisons of meltwater sustainability and is not a definite measure of the PSI surface mass balance.


We aggregate seasonal snow and annual mass loss statistics within the boundaries of discrete PSI features, with each feature >0.01 km$^2$ classified as a glacier (if crevasses are present indicating movement), perennial snowfield, rock glacier, buried ice, or semi-annual snowfield (Fig. 2). We identify glacier and perennial snowfield boundaries primarily from polygons delineated manually by Fountain et al. (2022; 2023). Due to the challenges of identifying buried ice and active rock glaciers

from aerial imagery, we supplement this dataset with additional features identified from spatially coherent patterns of elevation change (Sect. 2.1). We acknowledge that our elevation-based approach only identifies areas of buried ice and rock glaciers that are actively melting, but we are primarily interested in identifying PSI that is contributing net mass loss to streamflow. Several apparent fossil rock glaciers that do not show coherent elevation change are excluded from our analysis. The buried ice category includes debris-mantled glacier margins and ice-cored moraine remnants, some of which show

morphological indications of movement (Supplemental Fig. S6). We note that some rock glaciers and buried ice features also have a surface snow expression, and the separation of these categories from connected glaciers or snowfields is not definitive (Anderson et al., 2018). We follow the criteria of Fountain et al. (2023) to separate rock glaciers and fore-field connected glaciers based on the presence of an intervening topographic depression.



**Figure 2: Photos from the WRR study area showing examples of a glacier (A), a perennial snowfield (B), a rock glacier (C), and buried ice (D).**

Classification of PSI features among the various categories is also aided by opportunistic photos curated from numerous recreational mountaineering trips in the study area between 2015-2024 (Fig. 2). We also define a new category of "semi-annual snowfields" that exhibit net mass loss between 2019 and 2023, but which are not identified as PSI >0.01 km$^2$ in the 2015 orthoimagery used by Fountain et al. (2023). Some of these semi-annual features are present in both the 2019 and 2023 lidar data, while others appear to have melted completely over that time. The numerous remaining features that are too small for manual classification are grouped together as "small snow patches," varying in size from one grid cell (9 m$^2$) to 0.01 km$^2$.

The repeat lidar surveys largely cover the study watersheds. A few small PSI features (0.01-0.03 km$^2$) representing no more than 0.5% of the PSI area in any watershed are outside the extent of the 2023 lidar survey. Also, a portion of the Upper Green River watershed extends outside the extent of the 2024 snow lidar survey (Supplemental Fig. S1). We impute SWE depths at 3 m resolution for this area with a deep neural network trained on the lidar-based SWE map based on topographic metrics and Landsat fractional snow cover (U.S. Geological Survey 2024b); details are included in Appendix B. On the test





set, the imputation model achieves an $R^2$ of 0.68 for 3 m resolution SWE with approximately unbiased mean (-0.08 cm) and
median (+2.5 cm). We only use the imputed SWE values to estimate snow storage metrics in the Upper Green River
watershed, thus enabling comparisons with other gauged watersheds. All PSI features considered in this study are within the
lidar survey domain, and all supraglacial snow depths are directly measured by lidar.

## 3.4 Mass Loss Uncertainty

Seasonal snow variations contribute potential uncertainty to the 2019-2023 estimate of PSI mass loss. The 2019 USGS lidar
data was collected on August 17-18, about a month before the typical end of the ablation season. The lidar intensity shows
bare ice below 3500-3600 m on the large glaciers, consistent with late ablation season conditions. However, some snowfields
are larger than their minimum extent observed in prior 2015 imagery (Fountain et al., 2023), suggesting that some seasonal
snow may remain in the 2019 lidar data. The 2023 lidar survey was acquired after an early October snowfall, but coincident
fieldwork indicates that snow depths were on the same order of magnitude as the ground roughness (~5-15 cm at 3800 m,
above the median PSI elevation of 3650 m). Due to the small (14 cm) coregistration error between surveys, we expect the
largest source of uncertainty in the mass loss rate to derive from potential differences in the amount of snow present between
late August of 2019 and early October of 2023. We derive a conservative upper bound for this uncertainty based on the 2024
SWE data, assuming that no more than 10% of the seasonal snowpack remains at the time of the ablation-season lidar
acquisitions. The assumption that >90% of snowmelt occurs by late August is supported by streamflow timing, since
September-October streamflow represents only 9.3% of the average May-October volume across the five study watersheds.
Thus, regardless of whether there was more snow in August 2019 (remaining from the previous winter) or more snow in
October 2023 (accumulated in the prior week), uncertainty introduced by seasonal snow variability should not exceed 10%
of the total seasonal snowpack. We evaluate the sensitivity of our watershed mass loss estimates to this plausible range of
snow variability by adding ±10% of the 2024 SWE depth to each PSI grid cell.


We combine uncertainty arising from seasonal snow variability and PSI density estimates to derive a final PSI mass loss
uncertainty. Assuming the two uncertainty sources are independent, we combine the relative uncertainty from seasonal snow
(±10% of the 2024 SWE) and the assumed PSI density ranges (±0.06 or ±0.10 g cm$^{-3}$, Table 2) for each grid cell. The lower
bound on mass loss is derived from the lower density with 10% of the 2024 SWE subtracted from the mass loss, while the
upper bound on the mass loss rate is derived from the higher density with 10% of the 2024 SWE added to the mass loss.

## 3.5 Topographic Indices

We define four topographic indices related to snow accumulation and the energy available for melt: elevation, solar shading,
upwind angle, and upwind area (Fig. 3). Elevation is a proxy for air temperature, assuming that temperature lapse rates are
consistent over the relatively small PSI study area (~400 km$^2$ bounding convex hull). We calculate mean ablation-season
(July-September) sun exposure time at 30 m resolution (WhiteboxTools library: Lindsay, 2016) to account for the effect of



terrain shading on solar insolation, which is the most influential factor controlling energy balance heterogeneity across a glacier (Wolken, 2000; Oerlemans and Klok, 2002). Shading time is a coarse energy balance metric that does not capture the variable effects of cloudiness and seasonality (Olyphant, 1984) or longwave radiation from rock walls (Olyphant, 1986). Nevertheless, we expect topographic shading to positively correlate with snow persistence and reduce the PSI ablation rate

(e.g., Williams et al., 1972; Munro and Young, 1982; Delmas et al., 2014; Olson and Rupper, 2019; Florentine et al., 2020).

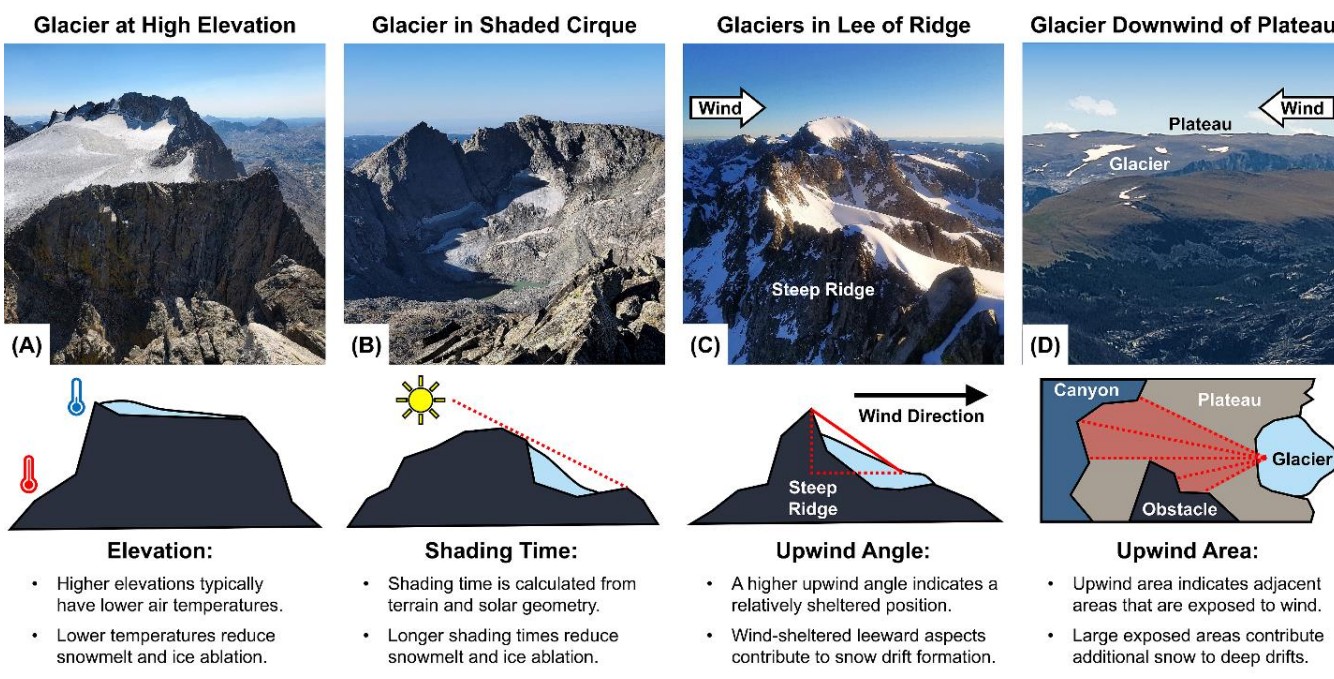

**Figure 3: Example photos and conceptual models of the four topographic indices for the glaciers shown in Fig. 7. The Upper Fremont Glacier (A, Fig. 7 A1-A5) is notable for its high elevation; the Stroud Glacier (B, Fig. 7 B1-B5) is notable for its shaded**
**position in a north-facing cirque; the Dinwoody Glaciers (C, Fig. 7 C1-C5) are notable for their position on the downwind side of a steep ridge, which provides wind shelter; and the glacier on Dry Creek Ridge (D, Fig. 7 D1-D5) is notable for the exposed plateau on its upwind side that provides a large drift source area.**

Topographic wind drift indices are also useful for investigating spatial patterns of snow accumulation and glaciation in mountain environments (e.g., Lapen and Martz, 1993; Purves et al., 1999; Winstral and Marks, 2002; Anderton et al., 2004;
Freudiger et al., 2017). We calculate the maximum horizon angle in the upwind direction (hereafter, the "upwind angle"), which is the Sx parameter defined by Winstral et al. (2002). Grid cells with a positive upwind angle are sheltered from the wind by terrain obstacles, which promotes drift formation, and a steeper upwind angle indicates a more sheltered location. Calculating the upwind angle requires defining a prevailing wind direction, which is approximately west to east based on anecdotal observations, continuous measurements by an automatic weather station in the Torrey Creek watershed (operated
on ranch land by the authors), and NLDAS-2 climatology (Supplemental Fig. S2; Xia et al., 2012). We calculate the upwind angle from a digital elevation model (Farr et al. 2007) at 30 m resolution with an unlimited upwind search distance.





Additionally, we calculate the same index using the 3 m lidar DTM with a maximum search distance of 100 m to capture small-scale terrain effects. We combine both resolutions by interpolating the coarser raster to 3 m and taking the maximum upwind angle from short- and long-distance calculations. Since wind varies between storms and is deflected by mountainous terrain, we average the upwind angle over an azimuth range from 240° to 300° in 1° increments to account for a range of plausible shelter and scour configurations.

A new topographic wind index, the upwind area, can highlight likely drift accumulation areas on the downwind margins of high elevation plateaus, which are a notable feature of the WRR (Anderson, 2002). Although the upwind angle is useful for explaining drift locations within small watersheds (e.g., Winstral et al., 2002), it does not account for variations in the size of contributing source areas, which is an important control on deep drift formation (Outcalt and MacPhail 1965). Elevated plateau surfaces in the northern WRR are conducive to long-distance snow transport, leading to deep drifts in downwind topographic depressions. Some of the drifting snow from these plateaus collects in cirque basins, but large drifts can also form in relatively shallow depressions where the upwind angle index does not predict drift formation (e.g., Figs. 3D and 7D). To better capture WRR drift patterns, we estimate the upwind area, a two-dimensional index based on the extent of upwind areas that are exposed to the wind and a downwind distance factor accounting for deposition and avalanches from steep cliffs (Appendix C Eqs. C1-C9). The upwind area index only considers potential contributing areas that are not separated from the local grid cell by large intervening sheltered areas, since terrain obstacles can interrupt wind transport (Fig. 3D).

To relate heterogeneity in PSI accumulation and ablation to wind drifting and energy inputs, we develop a classification based on the four topographic indices (elevation, shading time, upwind angle, and upwind area). The topographic indices are linearly rescaled within the interquartile range of topographic variability across all PSI grid cells. This approach to normalization preserves heavy distribution tails, which helps identify cells dominated by a given index. For grid cells with at least one topographic index exceeding the upper quartile of that index for all PSI cells, the index with the largest re-scaled value is assigned as the dominant classification. Aggregating SWE and mass loss rates within cells dominated by a given topographic index provides a measure of each index's relationship to accumulation and ablation patterns. To evaluate the relationship between wind drifting and seasonal snow at the watershed scale, we also classify snow-covered grid cells that exceed the median value of the upwind angle and/or upwind area indices within the full snow-covered area. We then aggregate SWE in areas with and without an above-median wind drift index using different minimum SWE thresholds.

## 3.6 Controls on Streamflow

Four USGS stream gauges each provide continuous daily flow records (Table 1; U.S. Geological Survey, 2024a). The Torrey Creek stream gauge is operated by the authors beginning with the 2022 water year. We impute Torrey Creek streamflow prior to 2022 using Gaussian Process regression (DiceKriging library: Roustant et al., 2012) based on the four other gauges and a seasonal sine wave. We test the validity of this imputation by predicting 2024 Torrey Creek streamflow from only the



2022-2023 data, which yields a daily Nash-Sutcliffe efficiency of 0.93 and a bias of 2%. We then use all available Torrey
Creek data (water years 2022-2024) to impute the 2005-2021 period, which is solely used to moderate the effect of discrete
storm events on the annual hydrograph for visualization and inter-watershed comparison purposes. We aggregate yearly,
August-September, and July-September mean streamflow for all five gauges over water years 2020-2023 to compare with
PSI mass loss between the 2019-2023 lidar surveys. Assuming that evapotranspiration is energy limited (but not necessarily

zero) along the downstream flowpaths from PSI areas as a result of ecosystem adaptation to perennial water availability
(e.g., Gentine et al., 2012), we can directly estimate the contribution of mass loss to streamflow, since an additional volume
of PSI melt would contribute an equal volume of additional streamflow. An energy limited assumption for the PSI
contribution to evapotranspiration is reasonable because meltwater from most PSI features immediately forms perennial
streams, demonstrating that the available water exceeds potential evapotranspiration along the downstream flowpaths.


We test the sensitivity of historic streamflow patterns to further PSI recession by subtracting watershed-average mass loss
rates from late-summer streamflow. We create daily average hydrographs for each gauge over the past 20 water years (2005-
2024) to reduce the effect of stochastic storms on our inter-watershed hydrograph shape comparisons, but we only consider
water years 2020-2023 to estimate the mass loss contribution since that period matches the lidar survey timing. Normalizing

each watershed's hydrograph by the mean May-September flow allows us to compare runoff timing, and normalizing
specific runoff (mm d$^{-1}$) by the daily mean across watersheds allows us to identify watersheds that produce relatively more or
less water at a given time of year. We estimate the potential impact of complete deglaciation on the downstream hydrograph
by reducing each watershed's August-September streamflow in proportion to the estimated PSI mass loss fraction. This
counterfactual scenario is a conservative lower bound on late-summer streamflow in the absence of PSI retreat because at

least some mass loss probably occurs earlier in the summer.

To identify dominant controls on streamflow patterns, we normalize melt-season hydrographs by area-averaged snow
storage metrics and identify periods where a particular metric best explains the streamflow differences between watersheds.
Streamflow normalized by snowpack volume is analogous to a runoff ratio. We also normalize streamflow by the volume of

water stored in areas with SWE deeper than the 75th, 90th, or 95th percentile across the five-watershed study area,
corresponding to a minimum depth of 0.65 m, 1.26 m, or 1.72 m of SWE. These dimensionless hydrograph metrics provide a
proxy for the relationship between snow persistence and streamflow production, since deep snow drifts melt relatively
slowly compared to the same snow spread thinly. For each day in May-September, we further normalize each of these
hydrograph metrics by the mean across watersheds to identify which watersheds are over- or under-producing streamflow

relative to a given snow storage metric. On a given day, we assume that inter-watershed streamflow patterns are best
explained by whichever normalized hydrograph metric has the lowest root-mean-square difference between watersheds. For
example, if streamflow divided by snow storage approximately equals a constant value for all watersheds at a given point in




the melt season, then we conclude that inter-watershed differences in snow storage can explain inter-watershed streamflow differences for that time period.

## 4 Results

The 2024 SWE map (Supplemental Fig. S5) shows a snow-covered area of 1144 km$^2$ within the five study watersheds on May 31 (51% of total area). The 2019-2023 elevation difference map (Supplemental Fig. S7) shows an area of 58 km$^2$ with elevation changes that are likely associated with snow and ice (Sect. 3.1). Over this period, 93% of all PSI areas exhibit a decrease in elevation and 7% exhibit an increase.

## 4.1 Mass Loss Estimates

We classify a total of 338 PSI features with area >0.01 km$^2$ (including semi-annual snowfields), all of which experienced a net mass loss (Table 3). The Sourdough and Grasshopper glaciers experienced a terminal retreat of 60-120 m between 2019-2023, the fastest retreat of all WRR glaciers (Supplemental Figs. S8-S9). Conversely, some PSI features did not exhibit appreciable area change despite thinning. Although the mean surface elevation of all PSI decreased, some glaciers and snowfields exhibited small areas of elevation increase, probably from shifting snow drifts. The mean annual mass loss rate for "large" glaciers (>0.5 km$^2$) is 27% faster than "small" glaciers. Relative to the mean across all glaciers (0.99 m yr$^{-1}$), the mean loss rate is 30% slower for perennial snowfields, 80% slower for rock glaciers, and 49% slower for buried ice.





| Classification | N | Area Range, Total Area (km²) | Annual Net Mass Loss Rate, 2019 to 2023 (m yr⁻¹) | | | | Total Mass Loss (km³ yr⁻¹) |
|---|---|---|---|---|---|---|---|
| | | | Min | Max | Mean | Std. Dev. | |
| Large Glaciers (>0.5 km²) | 12 | 0.55-2.31 14.8 | 0.83 (0.75-0.91) | 1.70 (1.54-1.85) | 1.19 (1.08-1.31) | 0.26 | 17.2x10⁻³ (15.5-18.9) |
| Small Glaciers (<0.5 km²) | 44 | 0.02-0.45 6.4 | 0.29 (0.24-0.34) | 1.60 (1.43-1.77) | 0.94 (0.83-1.05) | 0.28 | 6.2x10⁻³ (5.5-6.9) |
| Perennial Snowfields | 116 | 0.01-0.26 4.1 | 0.09 (0.04-0.15) | 1.33 (1.05-1.60) | 0.69 (0.54-0.85) | 0.21 | 2.9x10⁻³ (2.3-3.5) |
| Rock Glaciers | 10 | 0.04-0.39 1.7 | 0.01 (0-0.03) | 0.73 (0.63-0.83) | 0.20 (0.16-0.24) | 0.24 | 0.37x10⁻³ (0.31-0.43) |
| Buried Ice | 65 | 0.01-0.31 4.3 | 0.03 (0.01-0.05) | 1.17 (1.05-1.30) | 0.50 (0.43-0.57) | 0.24 | 1.9x10⁻³ (1.7-2.2) |
| Semi-Annual Snowfields | 91 | 0.01-0.40 5.2 | 0.11 (0.07-0.15) | 0.72 (0.54-0.90) | 0.32 (0.22-0.42) | 0.11 | 1.5x10⁻³ (1.0-1.9) |
| Snow Patches (<0.01 km²) | - | 0.00-0.01 21.2 | - | - | 0.20 (0.12-0.27) | 0.21 | 4.2x10⁻³ (2.6-5.8) |
| **Entire PSI Area** | **-** | **57.6** | **-** | **-** | **0.59** | **-** | **34.2x10⁻³ (28.8-39.7)** |

**Table 3: Mass loss rates for different PSI feature classifications. Area measurements represent the feature area in 2019. The area range is the minimum and maximum for the N features actually included in that class. The mass loss minimum, maximum, and mean are area-averaged within discrete features in that class. Values in parentheses are uncertainty bounds.**

The 12 large glaciers contribute 50% of the total mass loss despite only representing 26% of the total PSI area. Small glaciers (11% of PSI area) contribute 18%, perennial snowfields (7% of PSI area) contribute 8%, rock glaciers (3% of PSI area) contribute 1%, and buried ice features (7% of PSI area) contribute 6%. The remaining 17% of mass loss derives from semi-annual features (9% of PSI area) and snow patches <0.01 km² (37% of PSI area). Considering only features >0.01 km² and excluding the semi-annual snowfields, glaciers contribute 82% of the mass loss, perennial snowfields 10%, rock glaciers 1%, and buried ice 7%.

The manually classified glaciers, perennial snowfields, rock glaciers, and buried ice features account for 83% of the total estimated mass loss between August 17-18, 2019, and October 7, 2023. The remaining fraction derives from features that are <0.01 km² at the time of the 2015 imagery used by Fountain et al. (2023). Among these features, 91 are larger than 0.01 km² in the 2019 data, indicating that they increased in area between 2015 and 2019. However, all of these features still experienced an elevation decrease between 2019 and 2023, with some vanishing completely. These "semi-annual" features



contribute 5% of the total mass loss between 2019 and 2023. Some of the semi-annual features could represent seasonal
snow remaining in the August 2019 data, which illustrates how variations in seasonal snow persistence contribute to
uncertainty in our annual PSI mass loss estimate (Sect. 3.4). Finally, 12% of the total mass loss derives from snow patches
that are detected by the 2019 lidar classification but <0.01 km$^2$ in all considered datasets. These snow patches are too
numerous for individual classification, and they likely represent a combination of semi-annual and perennial features.

The total PSI mass loss uncertainty is ±16%, including uncertainty propagated from the density assumptions and potential
variability in seasonal snow between the lidar survey dates. Seasonal snow variability is a smaller fraction of the measured
elevation change for glaciers, which experience the largest changes (Table 3). Consequentially, mass loss from large glaciers
and small glaciers is less uncertain (±10% or ±11% respectively) than mass loss from buried ice (±13%), rock glaciers
(±16%), perennial snowfields (±21%), semi-annual snowfields (±30%), and small snow patches (±38%).

**4.2 Seasonal Snow and Net Mass Loss Relationships**

Enhanced snow accumulation in cirque basins can decouple spatial patterns of PSI net mass loss from elevation-driven
climate gradients. Figure 4 illustrates the difference between two glaciers that are endmembers of accumulation and mass
loss variability in the WRR. The example small glacier (0.21 km$^2$, unnamed but c. Ross Lake) has a 26% slower mass loss
rate compared to the example large glacier (1.47 km$^2$, Grasshopper Glacier), despite the large glacier's 300 m higher median
elevation. The small glacier has a mean 2024 SWE depth of 4.0 m, 160% more than the large glacier (1.5 m mean SWE). A
decoupling of the mass loss rate from elevation is also observed within the perimeters of these glaciers. Lower elevations
exhibit a consistently faster mass loss rate for the large glacier (Pearson correlation r = -0.70) with the exception of a lake-
terminating portion. In contrast, there is no linear relationship between elevation and mass loss rate for the small glacier (r =
-0.05). Instead, spatial variability in the small glacier's mass loss rate is positively correlated with SWE (r = 0.65), unlike the
large higher elevation glacier, which has a negative correlation between mass loss and SWE (r = -0.41). The positive
correlation between deeper snow and faster mass loss within the small glacier is unintuitive, and this relationship appears to
indicate that the small glacier is more sensitive to annual snow accumulation rather than temperature. Reduced accumulation
or reduced seasonal snow persistence in 2023 compared to 2019 likely accounts for some of the small glacier's mass loss.



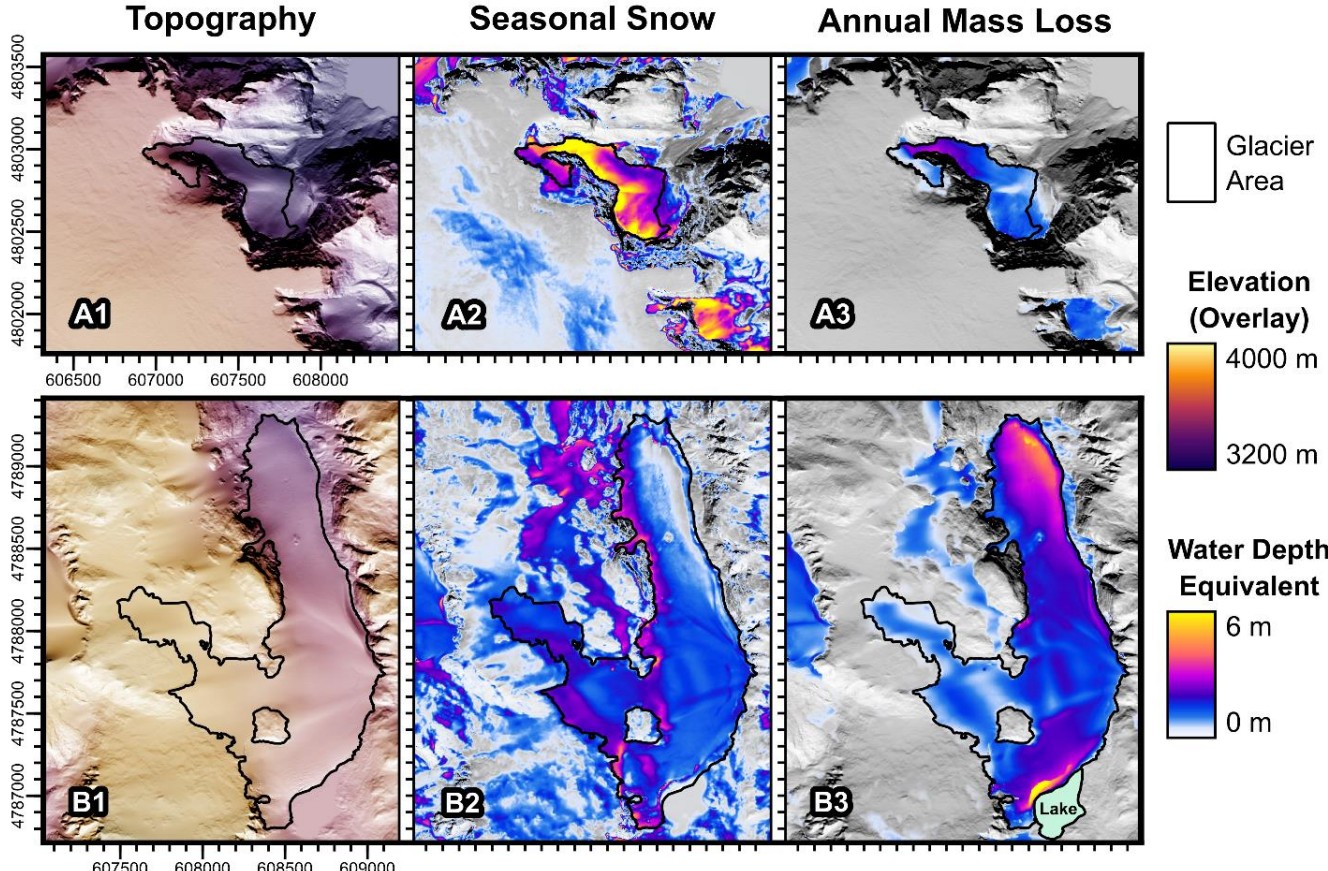

**Figure 4: Comparison of 2024 seasonal snow water equivalent (SWE) and 2019-2023 annual mass loss rate for a small glacier near Ross Lake (0.21 km², A1-A3) and the large Grasshopper Glacier (1.47 km², B1-B3). Glacier perimeters and areas date to 2019. A1 and B1 illustrate topography with shaded relief and a color-coded elevation overlay. A2 and B2 illustrate seasonal snow. A3 and B3 illustrate annual mass loss. Seasonal snow and annual mass loss are both in units of water depth equivalent with the same color ramp. Photos of these glaciers and their terrain indices are shown in Supplemental Figs. S15-S16.**

Comparing the hypsometry of seasonal snow measured in 2024 with the 2019-2023 mass loss rate across all PSI features classified as glaciers or perennial snowfields, we observe greater variability among features <0.5 km² (Fig. 5 and Supplemental Figs. S10-S12). The deepest snow accumulation zones generally occur at relatively low elevations. By only considering snow hypsometry within the perimeters of perennial features, we selectively sample areas of deep snow accumulation. At the watershed scale, deeper mean SWE depths are observed at relatively high elevations (Supplemental

Figs. S13-S14). The large glaciers (>0.5 km²) exhibit faster mass loss rates at lower elevations, consistent with the example large glacier in Fig. 4. Within the perimeters of the large glaciers, the mass loss rate has a mean correlation of -0.60 with elevation, in contrast to small glaciers, where the mean correlation is weak (r = -0.16).





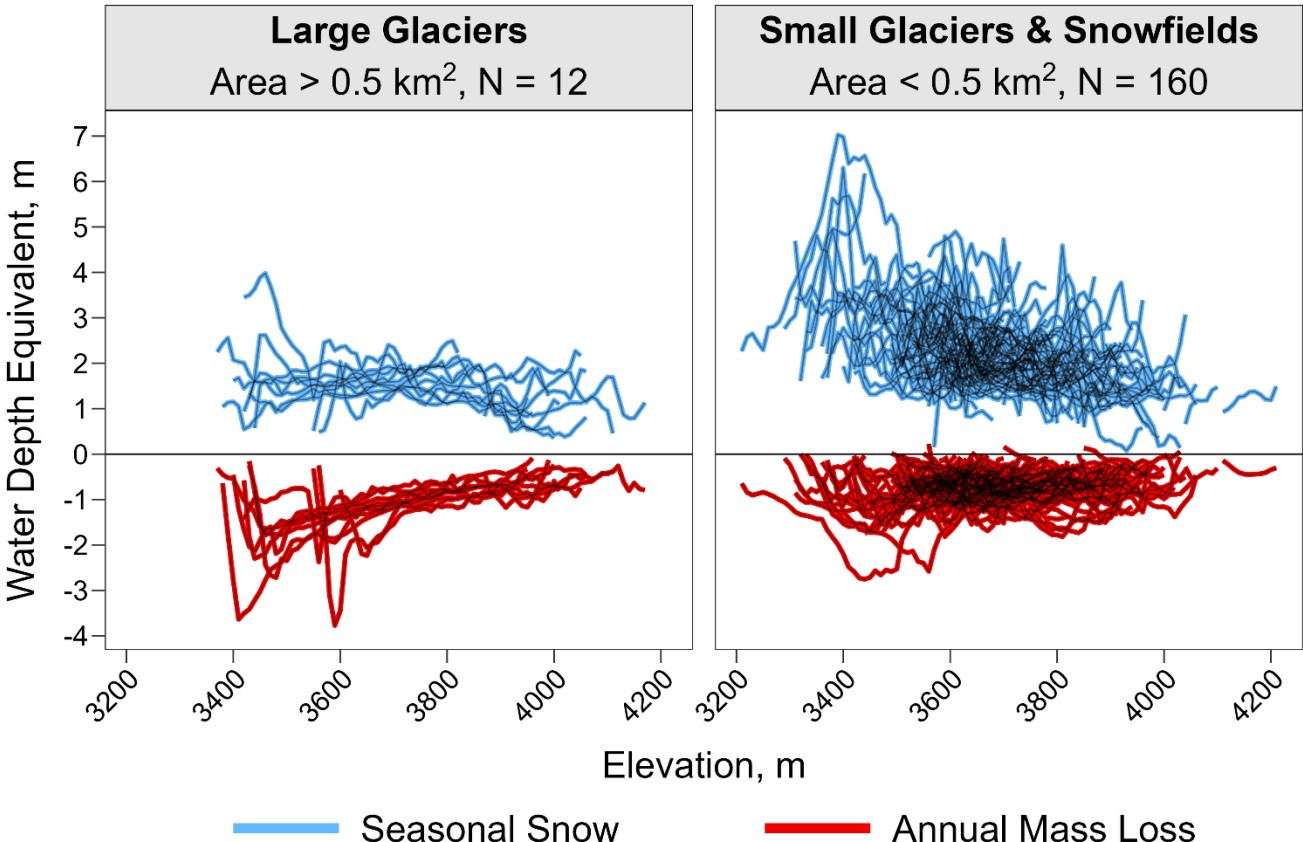

**Figure 5: Elevation hypsometry of seasonal snow (2024) and annual mass loss (2019-2023) for large glaciers or small glaciers and perennial snowfields, showing a wider range of variability in both accumulation and ablation for relatively small features. Example photos and individual hypsometry profiles from each of the large glaciers are shown in Supplemental Figs. S10-S11.**

Net mass loss contributes a smaller fraction of the total meltwater generated by PSI features with relatively deep seasonal snowpacks (Fig. 6). In general, the large glaciers have relatively shallow mean SWE depths (1.0-1.8 m), with a roughly equal annual mass loss rate (0.8-1.7 m). As a result, the mass loss contribution for the 12 large glaciers varies between 36-53%. The deepest snow accumulation zones sustain relatively small PSI features. For glaciers and perennial snowfields with more than 3 m of seasonal SWE (N = 29, area 0.01-0.21 km$^2$), only 10-26% of yearly meltwater derives from net mass loss. Perennial snowfields tend to have deeper seasonal SWE and a smaller mass loss contribution compared to glaciers (Fig. 6). Rock glaciers exhibit the shallowest snow accumulation (mean 1.0 m SWE). Nevertheless, rock glaciers have a mean mass loss contribution of 13%, much smaller than the mean mass loss contribution of 31% for glaciers and snowfields with comparable seasonal SWE <2 m. Buried ice features, including debris-mantled glacier margins, similarly have relatively low mean SWE (1.6 m), but the mean mass loss contribution from buried ice (25%) is almost twice as high as from rock glaciers.




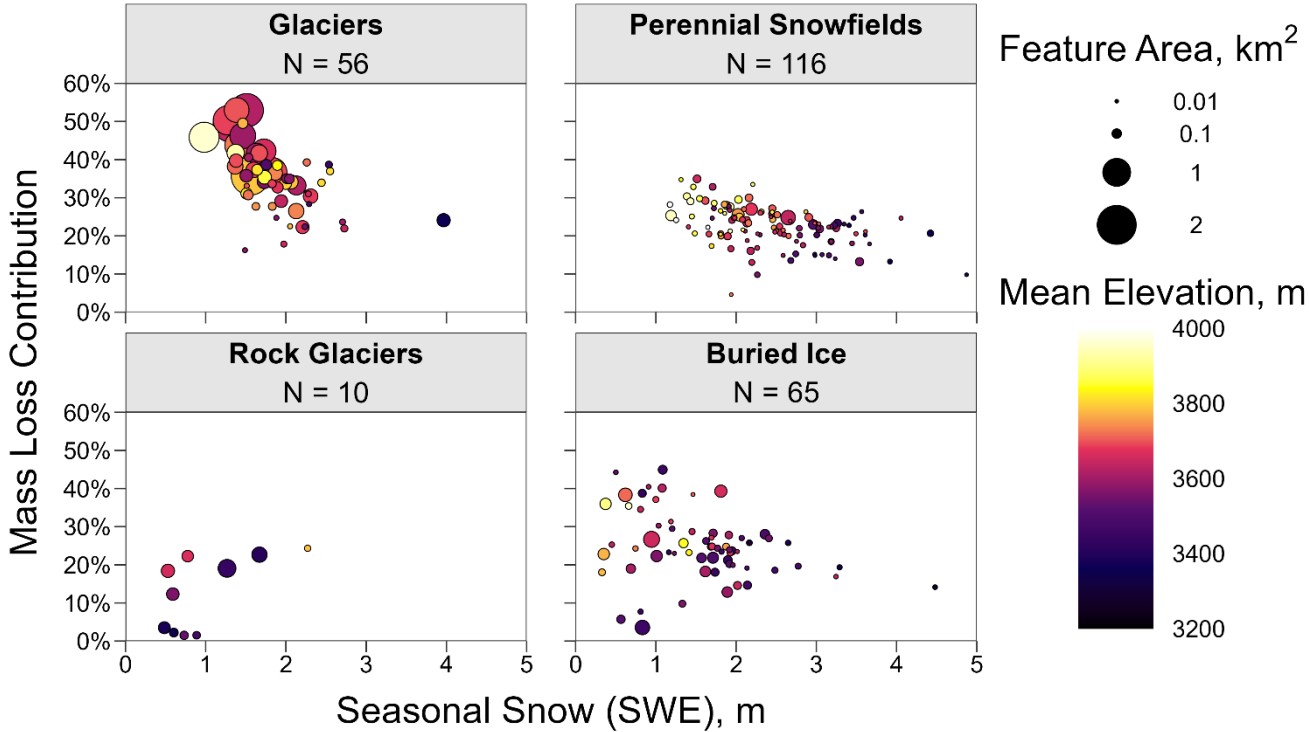

**Figure 6: Approximate fraction of total meltwater (seasonal snow plus annual net mass loss) that is derived from net mass loss for**
**247 PSI features classified as glaciers, perennial snowfields, rock glaciers, or buried ice. Photo examples of a glacier, perennial**
**snowfield, rock glacier, and buried ice are shown in Fig. 2.**

## 4.3 Topographic Controls

Indices related to wind drifting and energy inputs reveal topographic controls on the interrelationship between snow and

mass loss. In Fig. 7, we compare terrain indices and the mass loss contribution for the same four glaciers from Fig. 3, each of

which exemplify one of the four terrain indices (elevation, shading time, upwind angle, and upwind area). Maps of seasonal

snow and annual mass loss for these glaciers are show in Supplemental Fig. S17.





**Figure 7: Maps of glaciers that exemplify each of the four terrain indices: elevation (A1-A5), shading time (B1-B5), upwind angle (C1-C5), and upwind area (D1-D5). The mass loss contribution defines the ratio of net mass loss to total ablation (seasonal snow and mass loss). All maps are at the same scale. Example photos of these glaciers and conceptual models of the terrain indices are shown in Fig. 3.**



The elevation-dominated (air temperature) Upper Fremont Glacier (Fig. 7 A1-A5) is characterized by a high elevation (mean 3960 m), gentle slope (mean 11°), and the shallowest snowpack of all 12 large glaciers (mean 1.0 m). Despite its high elevation, this glacier experiences a relatively high mass loss contribution of 46%.


By contrast, a steep north-facing cliff shelters the Stroud Glacier (Fig. 7 B1-B5) from solar insolation. Despite being 400 m lower than the Upper Fremont Glacier, the Stroud Glacier is shaded for 71% of daylight hours and spends less than half as much time exposed to the sun, limiting its mass loss contribution to 34%.

A sharp north-south ridgeline contributes to wind drifting on top of the Gooseneck and Dinwoody glaciers (Fig. 7 C1-C5), which have steep upwind angles (mean 39° and 31°) providing wind shelter. These heavily drifted glaciers have about twice as much seasonal snow (mean 2.1 and 1.8 m) as the Upper Fremont Glacier despite their 180-270 m lower mean elevation.

Although both wind indices can identify potential drift formation regions in cirque basins (Supplemental Fig. S15 A2-A3),
the upwind area index also highlights shallow depressions downwind of large plateaus. The glacier exemplifying the upwind area index (Fig. 7 D1-D5, unnamed but c. Dry Creek Ridge) has a relatively low elevation (mean 3680 m), minimal terrain shading (mean 34% shade time), and moderate upwind angle (mean 17°), but it still has the smallest mass loss contribution (22%) of all glacier and perennial snow features with area >0.1 km$^2$. The upwind area index suggests that this glacier could accumulate snow from a ~3 km$^2$ contributing area. As a result of shifting snow drifts, this glacier exhibits an increase in
surface elevation over 8% of its area between 2019 and 2023 despite its 0.63 m yr$^{-1}$ mass loss rate.

The deepest snow is almost all associated with topographic controls on wind transport (Fig. 8). Although only 8% of all PSI grid cells have seasonal SWE >3 m, these deep accumulation zones are important for the sustainability of PSI meltwater because deeper accumulation is associated with a lower net mass loss contribution (Fig. 6). For PSI with SWE >3 m, 63% of
grid cells exceed the top quartile of wind drift indices (upwind angle or upwind area), and this fraction increases to 76% for SWE >4 m (2% of area) and 92% for SWE >6 m (0.2% of area). Large upwind contributing areas produces the deepest drifts, and the upwind area index is the dominant control for 68% of PSI grid cells with SWE >6 m.



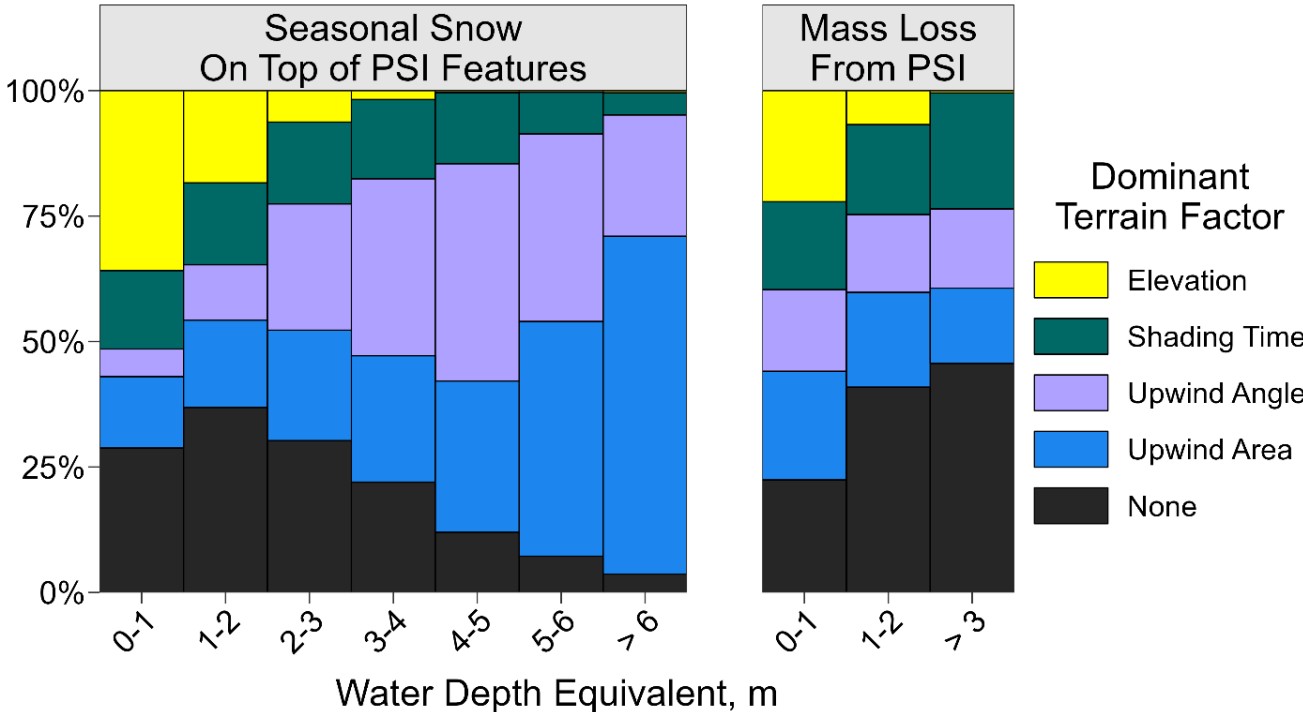

**Figure 8: Relative importance of terrain-based indices for PSI grid cells across a range of 2024 seasonal snow water equivalent (SWE) depths and 2019-2023 annual net mass loss rates.**

PSI features with elevation as the only dominant topographic factor (16% of PSI area) have relatively low snow accumulation and slow mass loss rates (Fig. 8). Elevation-dominated PSI features have a mean SWE depth of 1.3 m and a mean mass loss rate of 0.47 m yr$^{-1}$, compared to 1.8 m and 0.59 m yr$^{-1}$ for all PSI features. The relatively shallow snow accumulation associated with elevation-dominated PSI features is also consistent with our finding that the deepest snow occurs at relatively low elevations (Fig. 5). There is a fairly consistent 14-16% of PSI area dominated by topographic shading for all SWE depth ranges <5 m, but only 7% of PSI area is dominated by shading for SWE >5 m.

Watersheds with favorable topography for wind drifting exhibit extensive areas of deep snow despite having less total snow accumulation (Fig. 9). Watersheds on the east side of the Continental Divide (Torrey, Dinwoody, and Bull Lake Creeks) exhibit relatively low total snow storage, with 24% less area-normalized SWE above 3000 m relative to the west-side watersheds (Upper Green River and Pine Creek). This landscape-scale pattern of snow storage is consistent with orographic precipitation from prevailing west-to-east storm tracks, and may be exaggerated by sublimation from snow blowing over the crest of the mountain range. The east-side watersheds have greater SWE depth variability, with a standard deviation of 0.63 m for SWE depths at 3 m resolution, 35% higher than the west-side watersheds (standard deviation 0.47 m). The 90th and 95th percentile SWE depths are 1.37 and 1.87 m in the east-side watersheds, compared to 1.19 and 1.51 m in the west-side





watersheds. Consequentially, despite having less total SWE (-24%) relative to the west-side watersheds, the east-side watersheds have about the same (-7%) SWE deeper than 1 m and almost twice as much (+90%) SWE deeper than 2 m.

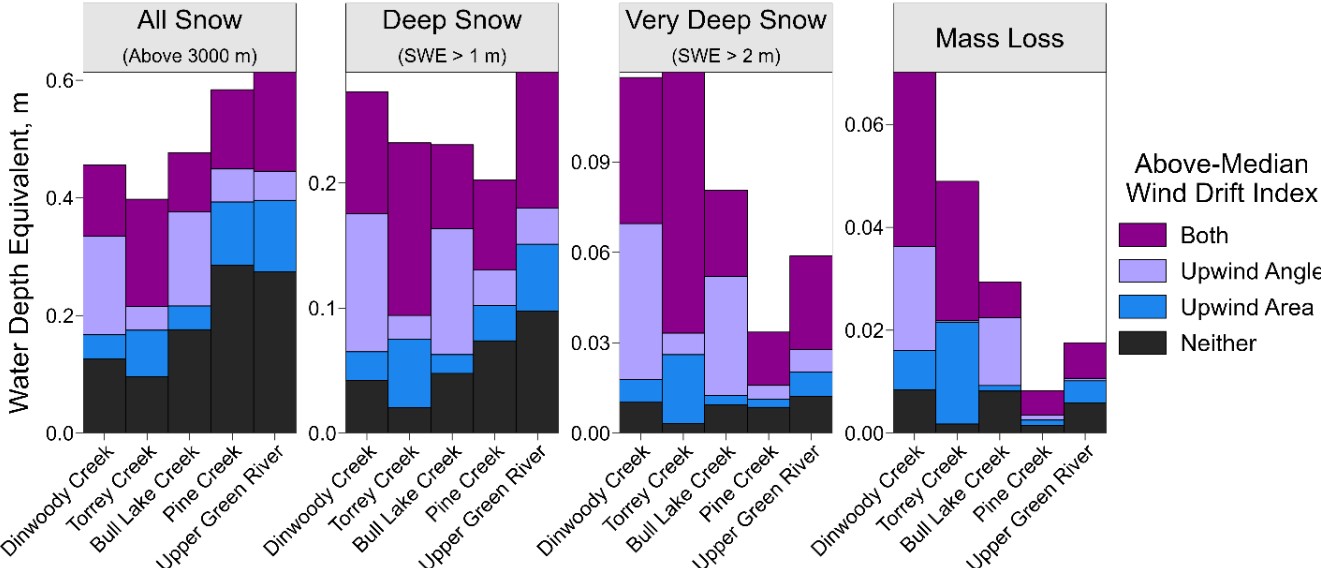

**Figure 9: Area-average 2024 SWE and 2019-2023 mass loss in each watershed, classified according to whether one or both of the topographic wind drift indices is above-median for the snow-covered area. Watersheds are sorted along the horizontal axis in order of decreasing PSI area fraction. For this comparison, we only consider areas above an elevation of 3000 m (approximately the snowline in the 2024 snow lidar survey) to control for the effect of stream gauge location relative to the contributing headwaters area.**

WRR watersheds with more total snow have relatively homogeneous spatial snowpack distributions, with much less deep snow and relatively few PSI features (Fig. 9). Across all five watersheds, there is a strong negative correlation (r = -0.90) between total SWE and SWE >2 m (both metrics area-normalized above the ~3000 m snowline). Watersheds with more deep snow (SWE >2 m) similarly have more area-normalized net mass loss (r = 0.94), opposite of the negative correlation between total snow storage and net mass loss (r = -0.79). In summary, the most heavily glaciated WRR watersheds have relatively less total seasonal snow, but the snow they do have is concentrated into deep drift zones, which are correlated with PSI abundance and net mass loss.

Spatial patterns of SWE depth are strongly related to wind drifting. Although it may appear from Fig. 8 that wind is not a dominant factor for areas with less than 2 m of SWE, recall that our PSI classification of topographic controls requires a grid cell to exceed the upper quartile of the respective terrain index, so 75% of the total PSI area is definitionally excluded from qualifying as "dominated" by any given factor. This more stringent classification criterion is useful to highlight differences between PSI features, but it is less applicable across the whole landscape. Considering the full 2024 snow lidar survey and not just PSI features, grid cells in the study watersheds with SWE >1 m (158 km$^2$) have a mean upwind angle of 21° and a mean upwind area of 0.96 km$^2$, 67% and 68% higher (more favorable) than snow-covered areas with SWE <1 m (776 km$^2$).





For areas with SWE >2 m (30 km$^2$), the mean upwind angle and upwind area increase an additional 32% and 31% respectively. By definition, half of all snow-covered grid cells are above-median for each topographic index, but areas with at least one above-median wind index represent 60% of the total SWE volume, 74% of SWE >1 m, and 87% of SWE >2 m (Fig. 9). The upwind area is particularly important in Torrey Creek due to high-elevation plateaus along the western (upwind) boundary of the watershed, while the upwind angle is relatively more important in Dinwoody Creek and Bull Lake Creek due to steep ridges that interrupt long-distance snow transport in those watersheds (Fig. 1).

**4.4 Streamflow Impacts**

PSI mass loss contributes a sizable fraction of late-summer streamflow in the five study watersheds (Table 4). Between August 17-18, 2019, and October 7, 2023, PSI net mass loss in the five study watersheds averaged 3.3x10$^{-2}$ km$^3$ yr$^{-1}$, equivalent to 3.6% of the total annual runoff over water years 2020-2023. Due to variable PSI abundance within each watershed, PSI net mass loss contributed ~1-2% of the annual water yield in west-side watersheds and ~4-9% in east-side

watersheds. Assuming that most net mass loss occurs in August and September, the PSI mass loss contribution to streamflow during that time ranges from ~10-14% in west-side watersheds and ~27-36% in east-side watersheds. Assuming a longer time period of July through September for the annual net mass loss, the average streamflow contribution during those three months is ~4-6% in west-side watersheds and ~12-18% in east-side watersheds. Uncertainty in the mass loss contribution varies between ±0.2-1.4% for the yearly contribution, ±1-3% for the potential July-September contribution, and ±2-6% for

the potential August-September contribution.




| Watershed | Area (km$^2$) | PSI Area Fraction | Mass Loss (km$^3$ yr$^{-1}$) | Mass Loss / Yearly Q | Mass Loss / Jul-Sep Q | Mass Loss / Aug-Sep Q |
|---|---|---|---|---|---|---|
| Dinwoody | 228 | 7.3% | 11.6x10$^{-3}$ (9.9-13.4) | 9.2% (7.8-10.5) | 18% (16-21) | 36% (31-42) |
| Torrey | 125 | 5.3% | 4.3x10$^{-3}$ (3.6-5.0) | 8.0% (6.7-9.3) | 16% (13-19) | 32% (27-37) |
| Bull Lake | 484 | 3.5% | 9.8x10$^{-3}$ (8.2-11.4) | 4.1% (3.4-4.7) | 12% (10-14) | 27% (22-31) |
| Pine | 196 | 1.8% | 1.3x10$^{-3}$ (1.0-1.6) | 0.9% (0.7-1.2) | 4% (3-5) | 10% (8-13) |
| Upper Green | 1212 | 0.9% | 6.2x10$^{-3}$ (5.3-7.1) | 1.7% (1.4-1.9) | 6% (5-7) | 14% (12-16) |

**Table 4: PSI mass loss contributions to streamflow for the five gauged study watersheds. Mass loss is estimated between 2019 and 2023, and streamflow (Q) is averaged over water years 2020 through 2023. Values in parentheses are the upper and lower uncertainty bounds.**

PSI mass loss cannot fully explain spatial patterns of late-summer streamflow among WRR watersheds. Normalizing seasonal streamflow hydrographs by the May-September mean flow for each watershed reveals variations in streamflow timing, and normalizing area-averaged daily flows by the mean across all watersheds stratifies area-normalized streamflow production at a given point in the season (Fig. 10). Watersheds with more PSI area produce relatively more late-summer streamflow (r = 0.73). The common explanation for this pattern is that ice melt from the glaciers contributes "extra" streamflow in watersheds with more abundant PSI (e.g., Fountain and Tangborn, 1985; Moore et al., 2009; Nolin et al., 2010; Clark et al., 2015). Indeed, the stratification of normalized July-September streamflow exactly matches the ordering of watersheds by PSI area fraction (Fig. 10B). Reducing the August-September streamflow from each watershed in proportion to the estimated mass loss contribution (Table 4) provides a lower bound on streamflow patterns in the absence of PSI recession. Watersheds with more abundant PSI still produce disproportionately more streamflow in the late summer, even assuming the yearly mass loss occurs entirely in August and September. Under recent historical conditions (2005-2024 mean), Dinwoody Creek (7.3% PSI area) and Torrey Creek (5.3% PSI area) respectively produce 129% and 75% more area-normalized August-September streamflow compared to the mean of the other three watersheds (PSI area 0.9-3.5%). In the lower-bound counterfactual scenario with no mass loss, Dinwoody and Torrey still produce 78% and 45% more area-normalized streamflow compared to the other three watersheds (Fig. 10C).



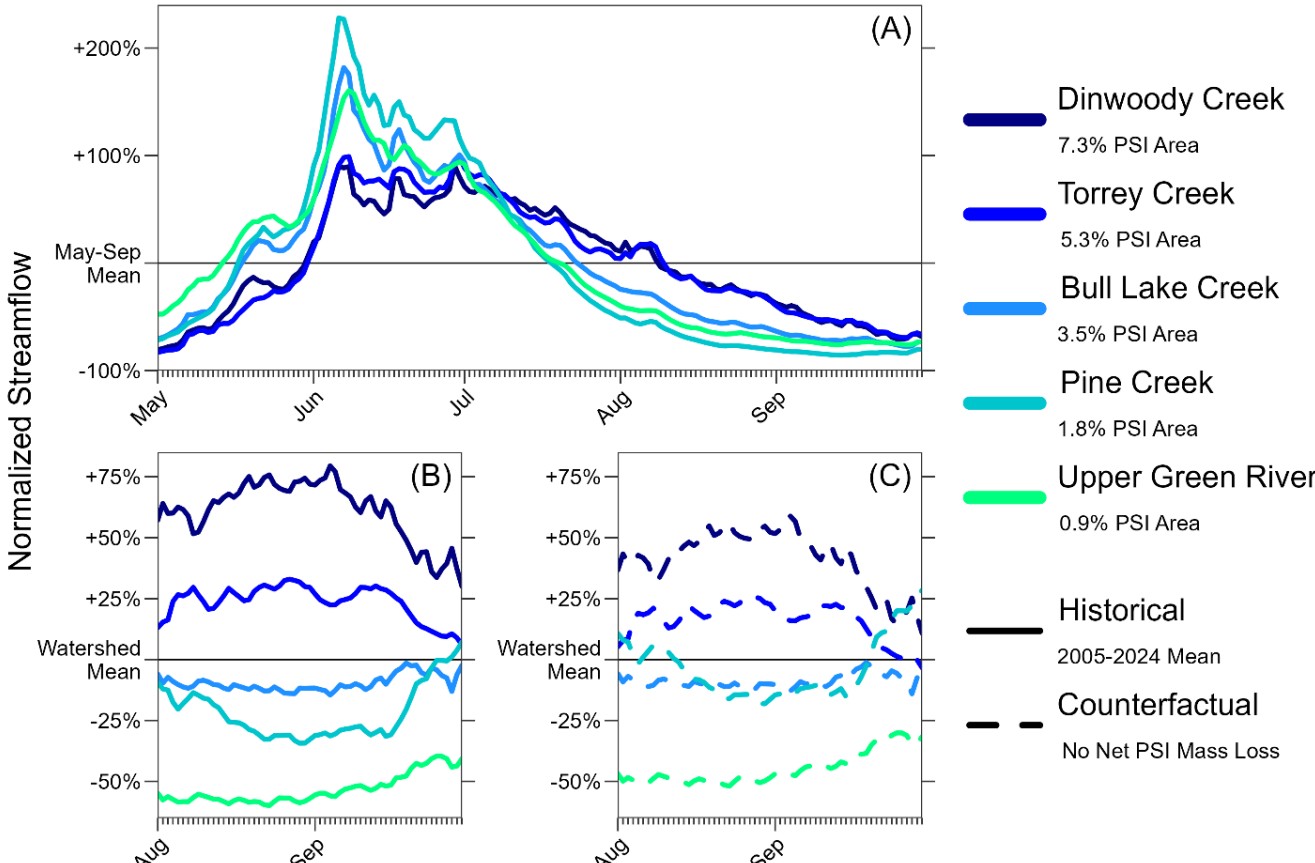

**Figure 10: Normalized daily streamflow hydrographs for the five gauged study watersheds (Fig. 1). Panel A shows historical streamflow normalized by the May-September mean in each watershed (independent of the other watersheds). Panels B-C show streamflow normalized by the daily mean specific streamflow (mm d⁻¹) across all watersheds. Historical streamflow (B) is averaged over the 2005-2024 period, and (C) is estimated by reducing August-September streamflow in proportion to the fractional contribution of perennial snow and ice (PSI) annual mass loss, assuming net mass loss is concentrated in those two months.**

Spatial patterns of deep snow drifts can largely explain differences in late-summer streamflow production between WRR watersheds. Considering only the snowmelt runoff contributing area above the ~3000 m snow line, there is a negative correlation (r = -0.60) between area-averaged snowpack storage and August-September streamflow, but there is a positive correlation between August-September streamflow and the snow volume stored in areas with SWE >1 m (r = 0.63) or SWE >2 m (r = 0.88). The negative correlation between total snowpack storage and August-September streamflow disappears (r = 0.08) when normalizing by the full watershed area instead of just the area above 3000 m due to the large separation between the contributing headwaters area and the Upper Green River stream gauge (Fig. 1). However, the correlation remains negative when considering just the other four watersheds, either when normalizing by the total watershed area (r = -0.62) or the contributing area above the snow line (r = -0.68).





The total snowpack volume best explains inter-watershed streamflow differences in the early melt season, from May through

early July, after which deeper snow depths provide better explanatory metrics for streamflow patterns (Fig. 11). Snow

storage in areas exceeding the 75th percentile (0.65 m) of all SWE depths on May 31, 2024, best explains inter-watershed

differences in area-normalized 2005-2023 streamflow during mid-July, SWE exceeding the 90th percentile (1.26 m) best

explains streamflow production from mid-July through early August, and SWE exceeding the 95th percentile (1.72 m) best

explains streamflow production from mid-August into September. Streamflow production near the end of the runoff season

is best correlated with SWE exceeding the 90th percentile, but this relationship may be confounded by watershed hypsometry

as temperatures begin to drop below freezing in late September. The streamflow patterns in Figs. 10-11 are robust to the

choice of time period and the imputation of streamflow data prior to water year 2022 in Torrey Creek; analogous patterns are

observed using only measured streamflow from water years 2022-2024, but the shorter averaging period introduces noise

due to stochastic storm events (Supplemental Figs. S18-S19).

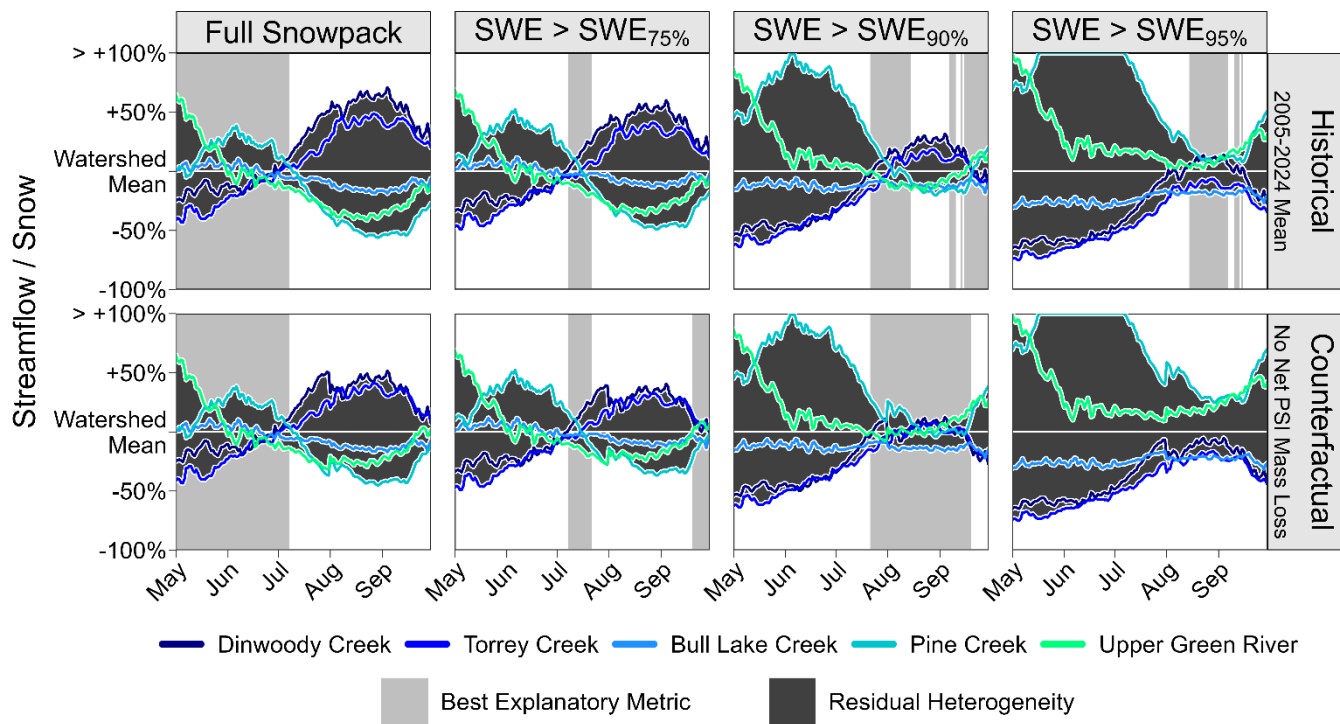

**Figure 11: Timeseries of dimensionless streamflow metrics normalized by the cross-watershed daily average. The upper row shows
historical streamflow metrics (2005-2024) and the lower row shows the same metrics assuming no net mass loss from perennial
snow and ice (PSI) features. Variability between watersheds (residual heterogeneity) indicates seasonal variations in the**
**relationship between snow storage and streamflow generation. Gray bars indicate which snowpack storage metric best explains
the inter-watershed variability in streamflow at different points in the season. $SWE_{X\%}$ = $X^{th}$ percentile SWE depth across all five
watersheds; in the May 31, 2024 data, $SWE_{75\%}$ = 0.65 m, $SWE_{90\%}$ = 1.26 m, $SWE_{95\%}$ = 1.72 m.**

The combination of PSI mass loss rate and deep seasonal snow storage best explains inter-watershed streamflow patterns.

Assuming no net mass loss, the >90th percentile SWE metric best explains inter-watershed streamflow variations for the full



period from late July through mid-September (Fig. 11). The relationship between historical (2005-2023) August streamflow and deeper snow storage (SWE depth >95$^{th}$ percentile) is likely a result of the confounding relationship between persistent seasonal snow and PSI abundance, which makes deeper snow metrics a partial proxy for net mass loss (Fig. 9). Reducing streamflow in proportion to net PSI mass loss (Table 3) prior to calculating the hydrograph metrics accounts for this correlation between mass loss and snow persistence. After accounting for net mass loss, we observe that residual differences

in late-summer streamflow generation among the watersheds (Fig. 10C) are approximately proportional to each watershed's snow storage in the deepest 10% of the seasonal snowpack (Fig. 11).

## 5 Discussion

### 5.1 Comparison of Mass Loss Contributions

The 2019-2023 lidar-derived geodetic elevation change data suggest an intermediate range of PSI mass loss rates between

faster and slower estimates from various methods over different time periods. Comparisons between PSI mass loss studies are complicated by interannual climate variability as well as methodological factors including the choice of watershed(s), the seasonal time period considered, and the definition of glacier meltwater (i.e., all runoff from glaciers or net mass loss only). Nonetheless, we compare our lidar-based mass loss estimates to prior studies based on area-volume scaling relationships, isotope unmixing, and meltwater runoff measurements.


Studies using area-volume scaling relationships may underestimate the mass loss contribution to streamflow on relatively short time periods. DeVisser and Fountain (2015) use multiple area-volume scaling relationships to estimate that mass loss was equivalent to 4-13% of the August-September flow in Dinwoody Creek and 2-6% in Bull Lake Creek between 2001-2006, with upper bounds about 3-5 times lower than the mean contribution estimated here (Table 4). However, on an earlier

period (1994-2001), the same study estimates a potential Dinwoody contribution as high as 23%, closer to our value of 36% for 2019-2023. Marks et al. (2015) estimate a relatively high mass loss contribution of 10-13% for July-September streamflow in Bull Lake Creek on the longer 1989-2006 period, close to our 12% estimate. Some of the variation in prior results may be attributable to episodic acceleration or slowing of PSI retreat, subjectivity in the delineation of boundaries, and the effects of variable image quality, resolution, and interpretation (Thompson et al., 2011; DeVisser and Fountain,

2015). Additionally, some PSI features show vertical thinning with minimal area change between 2019-2023, which could cause area-volume relationships to underestimate mass loss, though precise PSI boundaries are difficult to determine in 2019 and 2023 due to variable snow cover at the time of both lidar acquisitions.

Conversely, studies based on isotope unmixing and discharge measurements may be prone to overestimation of mass loss

contributions to streamflow if the contribution of supraglacial snowmelt is neglected. Cable et al. (2011) estimate a mass loss contribution to August streamflow as high as 70% in 2007 and 53% in 2008 for Dinwoody Creek, as much as 1.9 times



higher than our August-September estimate. Similarly, Vandeberg and VanLooy (2016) estimate a PSI mass loss contribution of 81-83% to Torrey Creek streamflow in August 2014 based on isotope unmixing from Continental Glacier meltwater. VanLooy and Vandeberg (2024) provide a lower estimate of 37% for August 2019 streamflow using similar
methods. These Torrey Creek estimates range from 1.2 to 2.6 times higher than the August-September contribution estimated here (32%). In the Bull Lake Creek watershed, VanLooy and Vandeberg (2019) estimate an August 2015 streamflow contribution of 56% based on direct measurements of discharge below major glaciers. This is 2.1 times higher than our August-September estimate here, but 2015 exhibited a particularly low summer snowpack with mostly bare ice exposed on the glaciers, potentially leading to a larger mass loss contribution to streamflow (VanLooy, personal
communication, October 2024).

Studies based on sampling meltwater below glaciers, whether for isotope analysis or by direct discharge measurement, may be affected by poor estimates of endmembers and potentially confounded by meltwater derived from lingering supraglacial snow that is not treated separately from multi-year mass loss. Notably, the lower Torrey Creek streamflow contribution
(37%) found by VanLooy and Vandeberg (2024) is based on ice samples from bare glacier ice, compared to the higher contribution (81-83%) based on meltwater at the glacier's terminus (Vandeberg and VanLooy, 2016), although melt and runoff rates also varied between the sampling years. The large contribution of total meltwater from PSI locations, as opposed to solely net mass loss from ice, is consistent with prior analyses (Bell et al., 2012) and supports our observation that PSI features coincide with persistent seasonal snow zones. Trace element measurements similarly support a distinction in the
sources of supraglacial meltwater and proglacial streams in the WRR (Barkdull et al., 2021).

**5.2 Spatial Heterogeneity**

Topography plays a crucial role in mediating snow persistence and annual mass loss in the WRR. With 338 distinct PSI features available for analysis, we are able to identify endmembers that exemplify particular topographic controls (Fig. 7). A simple linear classification of dominant topographic factors for the entire PSI area quantifies the importance of wind drifting
for deep snow accumulation (Fig. 8). The most hydrologically resilient WRR PSI features (low mass loss contribution) typically benefit from a combination of shading and wind drifting, which combine to create topographic refugia where accumulation is enhanced and ablation is limited (e.g., Supplemental Fig. S15, A1-A3). While we implicitly incorporate the role of avalanching from cirque headwalls in the upwind area index (Appendix C Eq. C6), we do not isolate the role of avalanches in this study, which may be particularly important for small, resilient PSI features (e.g., Laha et al., 2017; Onaca
et al., 2022). Spatiotemporal albedo variations also modulate the effect of shading and solar insolation on PSI mass loss (e.g., Wolken, 2000; Davaze et al., 2018; Owusu-Amponsah et al., 2022). Future research could explore additional processes controlling mass balance heterogeneity and investigate the interaction of topographic factors identified here through multivariate statistical analysis (e.g., Florentine et al., 2020), distributed field measurements (e.g., Hannah et al., 2000) or numerical simulations of topographic controls (e.g., Kessler et al., 2006).





Spatial heterogeneity in the PSI mass loss contribution requires caution in the extrapolation of field measurements to the watershed scale. Discrepancies between this study and some prior estimates of the mass loss contribution to streamflow may be partially explained by selective sampling of the largest, most notable glaciers (e.g., Marston et al., 1991; Vandeberg and VanLooy, 2016). Relatively large glaciers (area >0.5 km$^2$) exhibit faster mass loss rates (Table 3) and reduced variability compared to the full diversity of PSI features (Figs. 5 and 6). Complete spatial coverage by airborne lidar surveys can reduce uncertainty in the PSI mass loss contribution to streamflow by mapping all snow and ice storage changes, directly constraining spatial variability and accounting for numerous features smaller than the 0.01 km$^2$ threshold adopted for manual polygon delineation (Fountain et al., 2023). Between 2019 and 2023, PSI features <0.01 km$^2$ contributed 12% of all net mass loss in the study area (Table 3), suggesting that perennial snow patches that are too small for manual delineation could be an important source of meltwater.

About 6% of all WRR net mass loss derives from buried ice, which includes debris-mantled glacier margins excluded from polygon delineation (e.g., Fig. 2D and Fig. 1 of Fountain et al., 2023) as well as several ice-cored moraines (Supplemental Fig. S6). Detecting these buried ice features is a strength of our lidar-based approach to mass estimation. With spatially complete elevation change data, we can identify buried ice features wherever patterns of elevation change are spatially coherent and consistent with surface morphology. This change-based approach to PSI detection largely avoids the problem of deciding which areas to include in PSI net mass loss calculations, thereby respecting the glacier-rock glacier continuum (Anderson et al., 2018). Delineating buried ice features based on visible surface cover in areas with coherent elevation change supports the hypothesis that buried ice and rock glaciers are more resilient to the current warmer climate compared to nearby surficial features (Jones et al., 2019), with slower mass loss rates than glaciers and snowfields despite less snow accumulation (Fig. 6 and Table 3).

### 5.3 Interrelationship of Snow Drifts, Glaciers, and Streamflow

Watersheds with the most deep snow storage also have the most abundant PSI features and the largest mass loss contributions to streamflow (Figs. 8 and 10 and Table 4). While the total snowpack volume controls freshet magnitude as expected (Fig. 11), adjacent WRR watersheds with more total snowpack storage at the start of the melt season counterintuitively produce less late-summer runoff after controlling for gauge location (r = -0.60). Wind drifting explains this unexpected inverse relationship. Watersheds with favorable topography develop more abundant deep snow drifts (Figs. 8 and 9), which persist through the summer and explain inter-watershed streamflow variations later in the melt season (Fig. 11). Consequentially, late-summer streamflow production correlates positively with deep snow storage (r = 0.63 or 0.88 for SWE >1 m or >2 m), reversing the negative correlation with total area-normalized seasonal snow. The importance of deep snow for late-summer streamflow is well-recognized (Luce et al., 1998; Winstral et al., 2013; Brauchli et al., 2017; Le et al., 2023), but our WRR lidar data provide a unique opportunity to quantify the role of deep snow drift storage in shaping



hydrographs across multiple watersheds using purely data-driven methods (Figs. 9 and 11). Delayed snowmelt runoff from watersheds with deeper drifts can increase the climate resilience of downstream communities by mitigating the effects of
snow drought, which is especially important in areas like the WRR where artificial storage capacity is limited (McNeeley et al., 2018; Gordon et al., 2024).

Landscape-scale wind interactions with snow exert a large control on the WRR water balance. Topographic shelter and the size of upwind contributing areas are the dominant controls on deep snow accumulation in the WRR, from the grid cell scale
(Fig. 8) to the watershed scale (Fig. 9). A new snow drift index based on the extent of potential upwind contributing area shows that some PSI features in the WRR could collect snow from an upwind contributing area as large as several square kilometers, roughly 1-2 orders of magnitude larger than the area of the resultant drift zone. These kilometer-scale contributing areas are consistent with the estimation of Outcalt and MacPhail (1965) for a drift glacier in a similar environment. Concentration of drifting snow from a large contributing area leads to lidar-measured snow depths of 6-12 m
(~3-6 m SWE) over a contiguous area larger than 0.1 km$^2$ in some areas (e.g., Fig. 4 A2). In the WRR, blowing snow accumulated over extensive plateau surfaces (e.g., Fig. 7 D4) may decouple drift depths from more conventional topographic wind indices like the upwind angle of Winstral et al. (2002). Spatial precipitation fields in alpine mountains remain highly uncertain (Henn et al., 2018), and it is possible that the greater total snowpack storage in west-side WRR watersheds is entirely a result of orographic precipitation (Table 1). Nevertheless, the combination of more abundant deep drifts and
reduced total snowpack storage in downwind watersheds (Fig. 9) could suggest a potential effect of sublimation during wind transport in the more heavily drifted watersheds.

PSI abundance in the WRR is the result of hysteresis from legacy climate conditions that permitted glacier growth in sufficiently cold and snowy areas (Meier, 1950; Matthews and Briffa, 2004). As glaciers retreat, they recede into cirques and
can transition into perennial or seasonal snowfields. Throughout this transition, topographic controls on the mass balance become increasingly dominant (Florentine et al., 2018). In the WRR, the most hydrologically resilient glaciers and snowfields (low mass loss contribution) have relatively deep seasonal snow accumulation (Fig. 6), which is associated with topographic controls on wind drifting (Fig. 8). Paradoxically, watersheds with the most resilient PSI features may have a larger mass loss contribution to streamflow during late stages of deglaciation simply by virtue of retaining more PSI features
for longer. This relationship is evident in the positive correlation (r = 0.94) between watershed-average seasonal snow storage in areas with SWE >2 m and watershed-average net mass loss (Fig. 9). In contrast, the west-side (upwind) WRR watersheds supported much larger ice sheets than the east-side (downwind) watersheds during the Pleistocene glacial maximum, since orographic precipitation favors glaciation on the upwind aspect when the mountain range is below the ELA (Birkel et al., 2012). The greater extent of west-side ice sheets during glacial maxima could reduce topographic relief on the
upwind side of the mountain range (Birkel et al., 2012), intensifying the present-day disparity in topographic refugia and associated PSI patterns between upwind and downwind watersheds (Fig. 9).





Since perennial snow features coincide with the most persistent areas of the seasonal snowpack, the relationship between mass loss and streamflow is confounded by seasonal snowmelt. Watersheds with the largest fraction of PSI features by area

have the greatest potential for streamflow reductions post-glaciation (Table 4), but these same watersheds will likely continue to produce considerably more area-normalized late-season streamflow compared to the other watersheds as a result of deep seasonal snow (Fig. 11). Incorrectly assuming that WRR watersheds would all behave similarly without PSI mass loss leads to a 45-78% underestimation of August-September streamflow in the most-glaciated watersheds, even in the conservative post-deglaciation counterfactual scenario (Fig. 10C).


In the WRR, inter-watershed patterns of glaciation and streamflow are both mediated by the interaction of wind and snow with topography. Spatial patterns of streamflow generation (i.e., differences between watersheds) are relatively robust to a post-deglaciation scenario in the WRR (Fig. 10B-C) because of topographic factors that enhance snow accumulation and persistence (Figs. 8-9), which drives streamflow timing (Fig. 11). Wind transport of snow has been recognized as a dominant

control on watershed-scale patterns of WRR glaciation for the past eight decades (Baker, 1946), but quantifying this interrelationship at high resolution reveals that wind will likely remain a salient driver of seasonal snow persistence and streamflow patterns even in a future with fewer glaciers.

## 6 Conclusions

Glaciers are often conceptualized as a driver of hydrological differences between mountain watersheds, but they are also an

indicator of spatial heterogeneity in processes controlling snow persistence. Due to the causal relationship between snow persistence and glacier formation, the accumulation zones of mountain PSI features necessarily coincide with the most persistent areas of the seasonal snowpack. Inter-watershed patterns of glaciation and streamflow production can both result from topographic mediation of the underlying processes that drive snow persistence, such as wind drifting and shading. Our observations in the WRR suggest that some mountain glaciers are more a result, rather than a driver, of hydrological

heterogeneity. Consequentially, deglaciating WRR watersheds that currently exhibit elevated late-summer streamflow production will likely continue to out-produce nearby watersheds because seasonal snowpack dynamics control streamflow timing. Although climatic disruptions such as a rain-snow transition could reduce the role of snow persistence in shaping hydrographs, we expect inter-watershed streamflow patterns to remain especially robust in cold, windy regions like the WRR. Greater synergy between research efforts in mountain glaciology and snow hydrology is needed to further explore the

interrelationships between snow and ice storage cycles and forecast their response to climate change.





**Appendix A: Description of Empirical Snow Density Model**

Based on the graphical analysis of partial regression residuals as discussed in Sect. 3.2, we posit an additive model of bulk snow density driven by three independent factors:

$$\mu_{Density} = f_1 + f_2 + f_3 \tag{A1}$$

The first factor includes a constant offset and scales linearly with snow depth:

$$f_1 = c_1 + c_2 * Depth \tag{A2}$$

The second factor is a generalized logistic function of elevation, describing a smooth threshold (interpreted as the ripening elevation, near the 0° C isotherm) where snow densifies most rapidly:

$$f_2 = \frac{c_3}{1 + e^{-c_4 * \left(\frac{Elevation - 3000}{1000} - c_5\right)}} \tag{A3}$$

The third factor is a rectangular hyperbola that reduces density with increased canopy cover:

$$f_3 = c_6 * \left(1 - \frac{c_7}{c_7 + Canopy\ Cover}\right) \tag{A4}$$

We use Bayesian sampling to generate parameter estimates assuming a normal error distribution with standard deviation σ:

$$Density \sim normal\left(\mu_{Density}, \sigma\right) \tag{A5}$$

Median values of 100 Bayesian samples for each parameter obtained by fitting 39 regional density measurements:

$c_1 = 0.57$, $c_2 = 0.015$, $c_3 = -0.22$, $c_4 = 6.3$, $c_5 = 0.32$, $c_6 = -0.14$, $c_7 = 0.060$, $\sigma = 0.026$

Taking the derivative of Eq. A3 and substituting median parameter values, we note that the fastest rate of density change corresponds to an elevation of 3320 m, consistent with regional observations of snow temperature, which indicate an isothermal elevation of ~3300 m.


An interactive version of the empirical density model is available online at: https://www.desmos.com/calculator/n54vbqb2qc.

**Appendix B: Description of Deep Neural Network for SWE Imputation**

As explained in Sect. 3.3, we use a deep neural network implemented in PyTorch (Paszke et al., 2019) to impute SWE at 3 m
resolution for the portion of the Upper Green River watershed that is outside the lidar survey area (Supplemental Fig. S1).

The following data are used as prediction features:

1.    Elevation (3 m resolution; all DEM derivatives also at 3 m resolution unless indicated)

2.    North apparent dip  (combines slope and aspect)

3.    East apparent dip (combines slope and aspect)

4.    Topographic Roughness Index (mean absolute elev. diff. relative to surrounding cells)



5.  Topographic Position Index (elev. diff. relative to mean of surrounding cells)

6.  Topographic curvature at 3 m resolution

7.  Topographic curvature at 30 m resolution, interpolated back to 3 m

8.  Topographic curvature at 300 m resolution, interpolated back to 3 m

9.  Fractional canopy cover from RCMAP (Rigge et al. 2021), interpolated from 30 m to 3 m

10.  Landsat fractional snow cover (fSCA), mean of 16 cloud-free obs., Mar.-Oct., 2020-2024

11.  Landsat fractional snow cover (fSCA), same day as lidar data (May 31, 2024)

12.  Upwind angle (Sx, Winstral et al. 2002)

13.  Upwind area index (Appendix C)

14.  Mean annual sun exposure time (topographic shading)

The model structure consists of these 14 input features, a ReLU layer of 10,000 neurons, dropout regularization (p = 0.5), and nine subsequent ReLU layers of 100 neurons each, with a single final linear neuron predicting SWE. The model is trained using the Adam optimizer (Kingma and Ba, 2017) with a batch size of 256, a learning rate of $3 \times 10^{-4}$, and CUDA GPU acceleration. Data are split into halves based on a 1 km square grid to avoid spatial autocorrelation. Half of the 1 km square areas are used for training with $10^{6}$ random samples from within the training areas. The other half of 1 km squares are used for validation and test sets, with $5 \times 10^{5}$ random samples each. We implement early stopping if the coefficient of variation ($R^2$) on the validation set falls at least 0.01 below the maximum achieved $R^2$ for 10 epochs. In this case, early stopping ended training after 64 epochs, with a validation $R^2$ of 0.679 for SWE at 3 m resolution.

SWE depth is imputed using the trained model with the same 14 features. Snow presence is masked using a Random Forest classifier (Breiman et al., 2002) trained on the same 14 features with 200 decision trees. The out-of-bag error rate for the snow presence classification is 7%. The final $R^2$ is 0.683 for SWE at 3 m resolution on the test set.

**Appendix C: Definition of the Upwind Area Snow Drift Index**

Let (x, y) refer to the Cartesian coordinates of an arbitrary grid cell, and let ($x_i$, $y_i$) refer to the coordinates of a local grid cell for which we will calculate the snow drift contributing area. Let (r, θ) refer to the polar coordinates at distance r along the θ azimuth from a given grid cell.

First, we define the upwind fetch distance (Lapen and Martz, 1993) for a given cell, $D_{fetch}$, as the minimum distance r along the upwind azimuth θ from the coordinates (x, y) satisfying Eq. C1, where $S_{min}$ defines a minimum upwind angle:

$$D_{fetch}(x,y) \equiv min\{r \mid Elevation(r,\theta) \geq Elevation(x,y) + \tan(S_{min}) * r\} \qquad (C1)$$

$D_{fetch}$ is thus the minimum distance to a "meaningful" upwind terrain obstacle in the θ direction.





Next, we calculate $D_{fetch}$ for all cells along the azimuth θ from $(x_i, y_i)$ and apply a binary threshold such that an upwind cell is

defined as "wind-exposed" if its $D_{fetch}$ is greater than some minimum distance $MinD_{fetch}$:

$$Exposure(r,\theta) \equiv \begin{cases} 0, & D_{fetch}(r,\theta) < MinD_{fetch} \\ 1, & D_{fetch}(r,\theta) \geq MinD_{fetch} \end{cases} \qquad (C2)$$

Note that Exposure is defined along an azimuth $(r, \theta)$ from the local cell $(x_i, y_i)$, but $D_{fetch}$ as used in Eq. C2 is calculated

relative to the outlying coordinates of $(r, \theta)$, such that $(x, y)$ in Eq. C1 is obtained from $x = x_i + r * cos(\theta)$, $y = y_i + r * sin(\theta)$,

with r reset to 0 for all $(x, y)$ in Eq. C1.


Not all upwind exposed areas can contribute to snow accumulation at $(x_i, y_i)$, since intervening sheltered zones may trap

snow and prevent further transport. Sheltered zones are defined by a minimum continuous unexposed distance $MinD_{shelter}$.

We define a maximum contributing distance $r_{max}$ along a given azimuth based on the closest upwind sheltered zone, not

counting the local sheltered zone containing $(x_i, y_i)$. Thus, an upwind sheltered area interrupts the line search if and only if

the search has previously encountered at least one closer exposed area:

$$r_{max}(\theta) \equiv min \left\{ r \; \middle| \; \begin{array}{l} \int_r^{r+MinD_{shelter}} Exposure(r,\theta) * dr = 0 \\ \wedge \int_0^r Exposure(r,\theta) * dr > 0 \end{array} \right\} \qquad (C3)$$

In practice, the continuous integrals in Eq. C3 are replaced with a run length encoding of the Exposure vector derived from

the discrete grid cells along the upwind azimuth from $(x_i, y_i)$. Thus, $r_{max}(\theta)$ is the distance to the start of the first continuous

run of unexposed cells exceeding the $MinD_{shelter}$ length that is after the first run of exposed cells. An upper bound on $r_{max}$,

denoted as $MaxD_{contrib}$, may be useful to prevent unreasonably long-distance wind transport in some cases.

Finally, we integrate the potential contributing area within an upwind angular sector defined by $[\theta_{min}, \theta_{max}]$ and the maximum

contributing distance along each azimuth $r_{max}(\theta)$ from $(x_i, y_i)$. In polar coordinates, the area integral is given by:

$$Area = \iint dA = \iint r * dr \, d\theta \qquad (C4)$$

Counting only exposed areas (defined by Eq. C2) and adding definite integral bounds from definitions introduced earlier, we

arrive at the formula for the potential upwind contributing area:

$$Contributing \; Area(x_i, y_i) = \int_{\theta_{min}}^{\theta_{max}} \int_0^{r_{max}(\theta)} Exposure(r,\theta) * r * dr \, d\theta \qquad (C5)$$

Since snow transported from a more distant location is less likely to be deposited at $(x_i, y_i)$ due to a combination of

sublimation and upwind drifting, we weight the contributing area by the transport distance from the closest exposed zone to

$(x_i, y_i)$. Portions of the downwind length integral with a topographic slope, $Slope(r, \theta)$, greater than $S_{steep}$ are not counted

towards the transport distance because mechanisms other than wind (i.e., avalanching) can redistribute snow from steep



slopes. Thus, the drift deposition distance is defined using a helper function NonSteep(r, θ), which zeros out the length integral over steep slopes:


$$NonSteep(r,\theta) \equiv \begin{cases} 0, & Slope(r,\theta) \geq S_{steep} \\ 1, & Slope(r,\theta) < S_{steep} \end{cases} \tag{C6}$$

$$Deposition\ Distance(\theta) \equiv \int_{0}^{min\{r|Scour(r,\theta)=1\}} NonSteep(r,\theta) * dr \tag{C7}$$

A scale factor related to deposition downwind of the contributing area along a given azimuth is defined using an exponential function with a halving distance of $HalfD_{drift}$:

$$Deposition\ Factor(\theta) \equiv e^{\frac{\ln(2)}{HalfD_{drift}} * Deposition\ Distance\ (\theta)} \tag{C8}$$


The upwind area index is obtained by combining Eqs. C5 and C8:

$$Upwind\ Area\ Index(x_i, y_i) \equiv \int_{\theta_{min}}^{\theta_{max}} \int_{0}^{r_{max}(\theta)} \frac{Exposure(r,\theta)}{Deposition\ Factor(\theta)} * r * dr\ d\theta \tag{C9}$$

Free parameters required to calculate the upwind area index are tabulated as follows:

$[\theta_{min}, \theta_{max}]$   - the upwind azimuth range

$S_{min}$   - the minimum upwind shelter angle to define fetch distance

$MinD_{fetch}$   - the minimum upwind fetch distance to classify a given location as wind-exposed

$MinD_{shelter}$   - the minimum continuous unexposed distance that interrupts wind transport

$MaxD_{contrib}$   - (optional) an upper bound on the maximum wind transport distance

$S_{steep}$   - the minimum terrain slope above which downwind transport distance is ignored

$HalfD_{drift}$   - the exponential halving distance for downwind snow deposition

Values used in the present study are as follows:

$[\theta_{min}, \theta_{max}]$   - 240° to 300° (west ± 30°, conservatively large range obtained from weather data and observed drifts)

$S_{min}$   - 7° (obtained by testing several thresholds for agreement with observed drifts)

$MinD_{fetch}$   - 1,000 m (minimally sensitive: most cells have Fetch = grid res. or Fetch → Inf)

$MinD_{shelter}$   - 500 m (obtained from measurement of observed drift patterns)

$MaxD_{contrib}$   - 5,000 m (minimally sensitive: $r_{max}(\theta)$ is usually fixed by $MinD_{shelter}$ in Eq. C3)

$S_{steep}$   - 45° (obtained from observed drift locations relative to the slope of cirque walls)

$HalfD_{drift}$   - 500 m (obtained from measurement of observed drift patterns)

**Code and Data Availability**

Data and code necessary to reproduce the results and figures are archived at https://doi.org/10.5281/zenodo.14291096 (Boardman 2024). Lidar data products acquired commercially by Airborne Snow Observatories, Inc. are not included in this
archive due to contractual restrictions but are publicly available at https://data.airbornesnowobservatories.com/.

**Author Contribution**

ENB was the principal investigator for this project, developed the conceptualization, wrote the funding proposal, and lead project administration, data curation, methodology, software development, visualization, and writing. AGF contributed to the methodology, validation, and writing. JWB contributed to data curation, methodology, software, and writing. THP
contributed to funding acquisition and data curation. EGB contributed to data curation and software. LW contributed to data curation and writing. AAH contributed to data curation, methodology, visualization, writing, and supervision.

**Competing Interests**

Author ENB is the owner of Mountain Hydrology LLC, which contracted for data acquisition and partially funded ENB. Authors JWB, THP, and EGB have financial interests in Airborne Snow Observatories, Inc., which acquired the lidar data
used here. Authors LW and AAH received funding through a Mountain Hydrology LLC subaward to the University of Nevada, Reno.

**Acknowledgements**

We gratefully acknowledge a large in-kind contribution from Airborne Snow Observatories, Inc., which partially supported the acquisition of the 2023 glacier lidar data. We thank Jeffrey VanLooy and Gregory Vandeberg for helpful discussion and
informal manuscript review. We thank Anne MacKinnon and the Wind River Water Resources Control Board for helpful discussion of the regional water resources context. Regional WRR snow field surveys on the Wind River Indian Reservation were conducted by permission of the Office of the Tribal Water Engineer. Photos of the study area were opportunistically curated from the authors' personal archives from recreational mountaineering trips between 2015-2024.

**Financial Support**

Work presented here was partially funded by U.S. Bureau of Reclamation Award R24AC00025-00. Author ENB was additionally supported by the NSF Graduate Research Fellowship Program, Grant 1937966. Author AAH was partially supported by NSF EAR 2012188.



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
