# Peer review of "Wind and Topography Underlie Correlation Between Seasonal Snowpack, Mountain Glaciers, and Late-Summer Streamflow"

_EGUsphere, 2024_

## Referee Comment (RC1)

Boardman et al. assess the contributions of perennial snow and ice to seasonal streamflow in the Wind River Range using lidar scans in five watersheds spanning from 2019-2023. The experiment is well-executed and provides important advances regarding the contributions of depleting PSI features to streamflow. My comments focus mostly on the presentation and organization of the information within the manuscript. Some aspects could be improved by providing additional reader context and more explicitly stating how sections relate to the research questions.

General comments:

The science presented in this paper is sound and well-executed. However, there are instances where the reader would benefit from additional context in 1. Why the research is necessary/important, and 2. How a specific part of the paper relates to the overall goals of the investigation.

Some examples of this are in the methods section. For example, section 3.2 could be improved by adding an opening paragraph discussing the importance of snow density and the difficulties of measuring it (not sensed by lidar). Then, begin to describe the process of constraining snow density throughout the study site. While most readers may know this, it is important to zoom out and frame the importance of different sections.

The opening paragraph of section 3 lays out the general structure of the methods section. This paragraph could be expanded to explain *why* each part is carried out and how it relates to the goals of the study. Second, a flow chart type diagram could be useful in visualizing how each part of the study fits together.

Overall, the methods section could be improved by delineating the technical and conceptual methodologies. The methods section goes into density modeling, georeferencing of lidar data, streamflow imputation, and other highly technical details. These are important parts of the study, but the length of these technical sections makes it difficult to understand how all the pieces fit together. Some kind of reorganization of the more/less technical aspects could improve readability. Possibly just adding an opening sentence/paragraph about each section's importance prior to going into the details could suffice.

The results section faces similar difficulties as the methods. In general, improving opening sentences to ensure that the readers understand what the section is talking about, how it relates to the research questions, and how the figures fit into the story is key. The introduction does a great job of framing the investigation and defining research questions. In some cases, it is unclear how each of the results sections relates to the research questions. The opening paragraph of Section 4.3 is an example of when the authors do a good job framing the section - this should be emulated in other areas. This is addressed further in the line-by-line comments.

The advantages of understanding wind and topography effects on PSI loss could be stated more explicitly. To me, an advantage of this study is that PSI loss effects could be predicted based on wind/topo alone without needing the lidar surveys etc. Could this work be expanded on beyond

the wind river range? The impacts of this study could be more explicitly stated in the discussion section.

Line-by-line comments:

15: 'assess' may be more appropriate than 'conduct' for the lidar surveys.

23: What does 'favorable topography' mean? Suggest re-wording.

35: Temporal variability of what? Re-word for clarity.

46: May be helpful to define ELA for the readers.

52: Define lidar?

65: Suggest the hydrograph 'represents' as opposed to 'determines'

74-75: I'm not sure what space-for-time means here. Also, I'm not sure what the purpose of the last sentence of this paragraph is.

95: The wording for question three could be improved. Should it be: 'are *differences* in inter-watershed patterns primarily the result of…' Also 'topographically mediated' seems to be extraneous and makes the question harder to follow.

131: This sentence is a key driver of the study and should be further up.

202: Suggest adding an opening sentence mentioning why density is important.. simple, but helpful to the reader to zoom out a bit.

299: This paragraph does not follow well with the preceding paragraphs. Suggest moving or improving the opening sentence.

386-393: The opening of this paragraph which discusses the imputation of the Torrey Creek streamflow distracts from the main methods used in the paper. The important part of this section is how streamflow is used to interpret PSI mass loss, not the method for streamflow imputation.

431: Adding additional reader context here could be helpful. Something along the lines of: 'To understand the percent of streamflow contributions of PSI mass loss, we first must quantify PSI mass loss…' Also, explain why it is important that we quantify the differences for each type of PSI feature.

466: The opening sentence could be improved. What will you be discussing in this section? How does it relate to the research questions?

485: This paragraph is difficult to follow. Explain why you are comparing the hypsometry of snow/PSI. I am particularly unsure about the meaning of the sentence which starts with 'By only considering snow hypsometry within the perimeters…'

513: This opening is great. The explanation of why this section is important as well as what exactly is being analyzed in the following figure. This should serve as a model for other sections.

Fig. 7: The figure or the caption should contain the names of the PSI features as described in the text.

Figure 11. It is difficult to visualize differences between the top and bottom rows. Maybe a difference plot as well?

541: Hard transition. This could be a different subsection? Or at least an opening sentence saying: 'we now examine the relative roles of topographic features on snow distribution at different SWE depths'. It also could be helpful to frame why the depth analysis is important.

630: Is there a figure/table that relates to this statement?

---

## Author Response (AR1)

**Wind and Topography Underlie Correlation Between Seasonal Snowpack, Mountain Glaciers, and Late-Summer Streamflow**

Elijah N. Boardman, Andrew G. Fountain, Joseph W. Boardman, Thomas H. Painter, Evan W. Burgess, Laura Wilson, Adrian A. Harpold

**Response to Reviewers**

We thank the reviewers for helpful feedback, which we have addressed in a revision. In addition to minor edits, we have incorporated the following substantial improvements:

- Added a new figure (Fig. 2) providing a methodological flowchart to help clarify how different parts of the study are related.

- Added a new figure (Fig. 13) summarizing the key correlations introduced in the manuscript title, i.e., the correlation between late-summer streamflow and deep snow or mass loss from perennial snow and ice.

- Added a discussion section addressing groundwater, evapotranspiration, and other potential controls on streamflow differences.

- Added a discussion section addressing the transferability of our results to other regions.

- Added better transitions between text sections with topic sentences introducing each aspect and tying the different sections back to the research questions.

Our response to each of the two reviewers is uploaded separately as a reply to those comments.

The rest of this document shows a track-changes version of the revised manuscript.

**Wind and Topography Underlie Correlation Between Seasonal Snowpack, Mountain Glaciers, and Late-Summer Streamflow**

Elijah N. Boardman, Andrew G. Fountain, Joseph W. Boardman, Thomas H. Painter, Evan W. Burgess, Laura Wilson, Adrian A. Harpold

**Response to Reviewer Comment RC1**

We greatly appreciate the reviewer's helpful comments on this manuscript, which we have addressed in a revision. Our comments are interspersed into the review in blue.

Following our comments, we have attached a track-changes version showing our edits.

Boardman et al. assess the contributions of perennial snow and ice to seasonal streamflow in the Wind River Range using lidar scans in five watersheds spanning from 2019-2023. The experiment is well-executed and provides important advances regarding the contributions of depleting PSI features to streamflow. My comments focus mostly on the presentation and organization of the information within the manuscript. Some aspects could be improved by providing additional reader context and more explicitly stating how sections relate to the research questions.

We appreciate the reviewer's sentiment and have clarified the presentation and organization.

General comments: The science presented in this paper is sound and well-executed. However, there are instances where the reader would benefit from additional context in 1. Why the research is necessary/important, and 2. How a specific part of the paper relates to the overall goals of the investigation.

We have added more general topic sentences to most sections that tie back to the research questions.

Some examples of this are in the methods section. For example, section 3.2 could be improved by adding an opening paragraph discussing the importance of snow density and the difficulties of measuring it (not sensed by lidar). Then, begin to describe the process of constraining snow density throughout the study site. While most readers may know this, it is important to zoom out and frame the importance of different sections.

We expanded the introduction of density variations into its own paragraph as suggested.

The opening paragraph of section 3 lays out the general structure of the methods section. This paragraph could be expanded to explain why each part is carried out and how it relates to the goals of the study. Second, a flow chart type diagram could be useful in visualizing how each part of the study fits together.

We added a more general introductory sentence describing the methods as a whole, and we added a flowchart illustrating key methodological aspects of the study, as suggested (Fig. 2).

Overall, the methods section could be improved by delineating the technical and conceptual methodologies. The methods section goes into density modeling, georeferencing of lidar data, streamflow imputation, and other highly technical details. These are important parts of the study, but the length of these technical sections makes it difficult to understand how all the pieces fit together. Some kind of reorganization of the more/less technical aspects could improve readability. Possibly just adding an opening sentence/paragraph about each section's importance prior to going into the details could suffice.

We agree that sometimes the technical details can get in the way of understanding the bigger picture. However, the manuscript already has 3 appendices and 14 pages of supplemental material, so we are concerned that moving additional material out of the methods section might make the methods too vague.

To strike a balance, we have kept the detailed methods interspersed throughout, but we hope that the general introductory topic sentences and methodological flowchart can help remind the reader of the bigger picture.

The results section faces similar difficulties as the methods. In general, improving opening sentences to ensure that the readers understand what the section is talking about, how it relates to the research questions, and how the figures fit into the story is key. The introduction does a great job of framing the investigation and defining research questions. In some cases, it is unclear how each of the results sections relates to the research questions. The opening paragraph of Section 4.3 is an example of when the authors do a good job framing the section - this should be emulated in other areas. This is addressed further in the line-by-line comments.

We thank the reviewer for highlighting an example section that we can use as a model to clarify other sections. We have taken this advice and added bigger-picture introductions to the other methods sections that attempt to tie the results back to the research questions.

The advantages of understanding wind and topography effects on PSI loss could be stated more explicitly. To me, an advantage of this study is that PSI loss effects could be predicted based on wind/topo alone without needing the lidar surveys etc. Could this work be expanded on beyond the wind river range? The impacts of this study could be more explicitly stated in the discussion section.

We have added a discussion section (5.4) to more explicitly address the transferability of our framework. One consideration is that there is not a 1:1 correspondence between snow/PSI and wind/topography. Although we can show that most of the deep snow is associated with favorable topography for drifting (deep snow → implies wind-sheltered), there are lots of areas that are sheltered from the wind but lack deep snow for unknown reasons (wind-sheltered does NOT necessarily imply deep snow). Thus, we cannot really predict snow patterns (and hence the PSI mass loss contribution, which is calculated with reference to snow). However, we can still broadly extend our interpretation that "PSI features in topographic refugia (shaded, sheltered from wind) are more likely to persist and have smaller mass loss contributions to streamflow."

Line-by-line comments:

15: 'assess' may be more appropriate than 'conduct' for the lidar surveys.

Updated text to "we compare lidar surveys…" to better reflect methods.

23: What does 'favorable topography' mean? Suggest re-wording.

Re-worded to "topography that is conducive to wind drift formation."

35: Temporal variability of what? Re-word for clarity.

Re-worded: "…reduces the warm-season streamflow volume and increases the temporal variability of downstream water supplies."

46: May be helpful to define ELA for the readers.

Added "Equilibrium Line Altitude" definition.

52: Define lidar?

Added "light detection and ranging" parenthetical.

65: Suggest the hydrograph 'represents' as opposed to 'determines'

Agreed and changed.

74-75: I'm not sure what space-for-time means here. Also, I'm not sure what the purpose of the last sentence of this paragraph is.

Added explanation of space-for-time studies: "where nearby watersheds with fewer glaciers are used as a proxy for the future of deglaciating watersheds."

Also clarified that the persistence of cold-water stream species after deglaciation supports our contention that deglaciated watersheds are not necessarily identical with nearby watersheds that were not recently glaciated.

95: The wording for question three could be improved. Should it be: 'are differences in inter-watershed patterns primarily the result of…' Also 'topographically mediated' seems to be extraneous and makes the question harder to follow.

We have mostly adopted this suggestion to clarify the research question.

131: This sentence is a key driver of the study and should be further up.

We have moved the sentence about the importance of reducing PSI mass loss uncertainty into the introduction in the paragraph where we explain that net mass loss can contribute a considerable fraction of streamflow in alpine regions.

202: Suggest adding an opening sentence mentioning why density is important.. simple, but helpful to the reader to zoom out a bit.

Agreed, added new introductory sentence.

299: This paragraph does not follow well with the preceding paragraphs. Suggest moving or improving the opening sentence.

We moved the paragraph about the lidar imputation into the lidar section (2.1).

386-393: The opening of this paragraph which discusses the imputation of the Torrey Creek streamflow distracts from the main methods used in the paper. The important part of this section is how streamflow is used to interpret PSI mass loss, not the method for streamflow imputation.

Agreed; we have moved the imputation discussion to a subsequent paragraph after the primary motivation for considering streamflow is introduced.

431: Adding additional reader context here could be helpful. Something along the lines of: 'To understand the percent of streamflow contributions of PSI mass loss, we first must quantify PSI mass loss...' Also, explain why it is important that we quantify the differences for each type of PSI feature.

We have added a better introduction to the mass loss results section addressing both points.

466: The opening sentence could be improved. What will you be discussing in this section? How does it relate to the research questions?

The introduction to the seasonal snow vs. mass loss results section has been improved to explain why it matters for the research questions about controls on the water supply.

485: This paragraph is difficult to follow. Explain why you are comparing the hypsometry of snow/PSI. I am particularly unsure about the meaning of the sentence which starts with 'By only considering snow hypsometry within the perimeters...'

We revised the opening of the paragraph about hypsometry to explain that we are using elevation as a proxy to investigate the balance of topographic and climatological controls on different features.

Additionally, we expanded the sentence about selective sampling to explain that the unintuitive finding of deeper snow at lower elevations is due to a "survivorship bias" where only the deepest snow areas qualify as PSI at low elevations.

513: This opening is great. The explanation of why this section is important as well as what exactly is being analyzed in the following figure. This should serve as a model for other sections.

We appreciate this feedback and have tried to provide similar introductory statements elsewhere.

Fig. 7: The figure or the caption should contain the names of the PSI features as described in the text.

Agreed; glacier names have been added to the figure.

Figure 11. It is difficult to visualize differences between the top and bottom rows. Maybe a difference plot as well?

We agree that the two rows of Fig. 11 (now Fig. 12) are similar, but we believe that adding a difference plot would be unnecessarily difficult to interpret without adding much to the narrative. Instead, we believe that the vertical gray bars ("Best Explanatory Metric") adequately highlight the salient differences between the "Historical" and "Counterfactual" scenarios, i.e., >95$^{th}$ percentile SWE is only the best metric prior to accounting for the mass loss (gray bar is present for historical scenario, not in counterfactual scenario).

541: Hard transition. This could be a different subsection? Or at least an opening sentence saying: 'we now examine the relative roles of topographic features on snow distribution at different SWE depths'. It also could be helpful to frame why the depth analysis is important.

We believe that this section dealing with topographic controls on different SWE depths is best included in the current section on Topographic Controls. However, we have improved the transition to this paragraph by stating that we now consider how the indices generalize beyond the four glaciers considered in Fig. 7.

630: Is there a figure/table that relates to this statement?

Yes, we have added a reference to Fig. 12, which supports the statement that deep snow patterns can explain inter-watershed differences in late-summer streamflow.

**Wind and Topography Underlie Correlation Between Seasonal Snowpack, Mountain Glaciers, and Late-Summer Streamflow**

Elijah N. Boardman, Andrew G. Fountain, Joseph W. Boardman, Thomas H. Painter, Evan W. Burgess, Laura Wilson, Adrian A. Harpold

**Response to Reviewer Comment RC2**

*We greatly appreciate the reviewer's helpful comments on this manuscript, which we have addressed in a revision. Our comments are interspersed into the review in blue.*

*Following our comments, we have attached a track-changes version showing our edits.*

This paper presents a thorough set of measurements and calculations to estimate contributions of permanent snow and ice (PSI) features versus seasonal snowpack contributions to late season streamflow. It is a long paper, more monograph in scope than is common in recent years. Kudos are deserved for assembling so much information bearing on an important question for this region. There are some novel ideas and analyses presented as well.

*We thank the reviewer for the kind words, and we recognize that the paper is quite long—and hope to partially offset the length by further distilling the take-home messages.*

While it is great to see all the information brought together to be able to more thoroughly understand it, the document did not feel particularly coherent at times, with deep dives into details (agreeing here that they are important details) making it difficult to see where the arguments and calculations are going. It's all there, but the thread that keeps the reader's attention clear on why they are reading the particular paragraphs and sentences is lacking. This is only offered as feedback for the authors and whether they want potentially many other readers to experience the article in this same way.

*To better show how all the details fit into the bigger picture, we have added a methodological flowchart (Fig. 2) that can hopefully help provide a framework for tying the various analyses back to the core topics: (1) quantifying snowpack and mass loss, (2) explaining variability based on topography, and (3) estimating streamflow impacts.*

There may be some choices that add to the challenge of the long paper with many details that the authors may wish to reconsider. For example, in at least two places, there is presentation of data using two different sets of PSI features. One set is just larger features with relative permanence, e.g. buried ice, rock glaciers, perennial snowfields, and glaciers (as shown in in Figure 2). Then there are additional smaller features grouped as "small snow patches" and "semi-annual" features. Choosing one of these might be helpful as it can become difficult to know which PSI set is being invoked in any given analysis/figure later in the paper, although sometimes both are. It's just one less thing that the reader needs to track. There are layers of classification schemes, so it might be helpful to think about how many are necessary for the primary messages of this document.

While we agree that the different types of PSI features add to the overall complexity, we believe it is important to include all of these types since they each contribute to the total mass loss between 2019-2023, and we are interested in this total mass loss contribution to streamflow.

However, we appreciate that it can be confusing which features are being invoked at different times. Thus, we have clarified that all types of features (glaciers, perennial snowfields, rock glaciers, buried ice, semi-annual, snow patches) are included in any results or figures referring to generic "PSI," and the specific categories are only treated separately when explicitly mentioned.

Further, we added two photos to Fig. 2 (now Fig. 3) so that it now shows all 6 types of PSI features under consideration.

In section 4.3 a key question that seemed to be approached from multiple angles, but not quite answered directly, is whether the PSI and deep snow are essentially in the same places. I could see this being addressed in a straightforward way either by giving a CDF of SWE across the whole area in a watershed compared to a CDF for the PSI areas or by plotting a logistic regression of probability of being a PSI pixel versus SWE. Maybe this would be appropriate to do by type of PSI, as only certain types may be correlated with greater depth.

We address this question at the watershed-scale in the present study. In many cases, it is true that "PSI and deep snow are essentially in the same places" (e.g., Fig. 5A). However, some of the larger glaciers have clearly separated accumulation and ablation zones, with little to no supraglacial snow accumulation in the ablation zone (e.g., Fig. 5B). Thus, our conclusion linking deep snow to PSI abundance is only valid at the watershed scale, and a smaller-scale analysis would need to account for the effects of glacier flow.

We expanded our discussion (Section 5.3) to address this idea.

Closely related, is that there is some discussion on lines 566 and 567 where sentences are made about differences of snow in classes of all, >1 m, > 2m. It might be worthwhile just to plot the cumulative density function of SWE in the 5 watersheds and indicate which are west and east.

Since there is comparatively little deep snow relative to the total snowpack volume (compare vertical axis scales in Fig. 10), cumulative density functions of SWE depth would over-emphasize variations on the relatively shallow end of the scale. We believe that Fig. 10 adequately shows the differences in deep snow between watersheds, and adding an additional figure with essentially the same information is undesirable given the manuscript length.

Figure 10 in section 4.4 seems to be attempting to demonstrate that streamflow differences reflect a similar message as in section 4.3, and I'm wondering how strong the evidence from this analysis is. Most of that argument seems to be encapsulated in Figures 10B and 10C and it is a little difficult to follow the details of the argument. 10B just looks to be a daily recentering and rescaling of flow in each watershed relative to the mean for each watershed during Aug and Sep, where the rank is Dinwoody, Torrey, Bull Lake, Pine, and upper Green, which matches the ranking in percentage in PSI area. Important to the argument, this ranking changes little if the loss from the PSI is removed from the Aug-Sep flow totals. What is left unclear is what produces the difference in runoff across watersheds. For instance, unit-area runoff from Dinwoody is the second highest rank annually, and unit area runoff from the Upper Green is lowest annually as well. Only figure 10A lets you know that there is a pattern in timing difference (essentially pdfs of flow timing), but the ranking in Aug-Sep in 10A does not match the ranking in 10B. There is a decent negative correlation between unit-area runoff and basin area for Yearly, Jul-Sep, and Aug-Sep periods. There is a similar but slightly weaker negative correlation between PSI fraction and basin area. A story that one could tell is that most of the snow accumulates in the higher elevation areas, and as basin size increases, the runoff generated from those higher elevation areas is diluted when reporting unit-area runoff, whatever time of year you are looking at. That is clearly not the whole story, but it is one more dimension of watershed difference that could be brought up to explain differences besides focused SWE accumulation and PSI. I would suggest a tighter argument for this section.

We have re-made Fig. 10 (now Fig. 11) incorporating the reviewer's feedback.

Panel A is unchanged, since we believe that showing the streamflow timing variations is easily interpretable and sets up the context for understanding differences between the watersheds.

Panels B-C have been replaced with bivariate scatterplots (with correlation info) relating the August-September mean streamflow to the fraction of the watershed covered by PSI in 2019.

Our key conclusion is that watersheds with more PSI would have substantially higher late-summer streamflow even without net mass loss. This is now directly illustrated by Fig. 11C, which shows that the ranking of August-September streamflow changes little even after subtracting the net mass loss entirely from these two months.

We have re-arranged and expanded the last paragraph of Section 4.4 to explain how the inter-watershed streamflow variability remains similar even with different normalization approaches. In particular, we can reach basically the same conclusion whether streamflow is normalized by itself (timing dependence only, Fig. 11A), normalized by watershed area (Fig. 11B-C), or normalized by the total snowpack (Fig. 12).

It is also not clear what the reader gets from seeing a time series in 10B and 10C, when the seasonal average could be used (all lines are slightly curved but nominally flat relative to the interbasin variability), either in a table, or in a single bivariate plot of the mean values for each stream from 10B and 10C plotted against each other (to show that there is at least rank correlation).

Agreed—we have updated 11B-C to a scatterplot showing the similar ranking.

In a related vein, the word "groundwater" does not appear in the paper. I've spent some time in the wind rivers, and there is indeed a great deal of old crystalline rock, but there is also a fair bit of sedimentary rock within the basins outlined by the gaged watersheds, substantial fracturing of the Precambrian batholith, and some famous and productive springs. The subject of within-season timing delays driven by transit time through groundwater has been popular in the hydrology literature in recent years (see e.g. Somers and McKenzie 2020 for some review in the context of mountains, snow, and glaciers). Is there any evidence or signal from the hydrographs that could clearly separate groundwater storage effects from snowpack storage? Again, I'm not doubting that differences in snowpack heterogeneity/PSI are a primary driver, but in the spirit of thoroughness and honoring a prevalent hypothesis within the literature, a short paragraph to acknowledge this perspective might be useful.

We thank the reviewer for pointing out this oversight in our discussion. We have added a new discussion section (third paragraph of Section 5.3) addressing groundwater and other factors as potential additional drivers of inter-watershed streamflow variability.

Figure 11 may address some of the problem seen in figures 10B and 10C, as the area is in both numerator and denominator, so it just describes the ratio of runoff on a particular day to total amount of snow above some threshold. The problem is that the only interpretation that we draw from it is that differences across the basins in the metric plotted (full snowpack and three other thresholds) shift in time as the threshold SWE divisor shifts. The initial broad grey bar in the "Full Snowpack" plot is not convincing, as +/- 50% is not all that equal in value. Consequently, the notion that "total snowpack volume controls freshet magnitude" (line 749) is not well supported. The freshet magnitude may be more related to the elevational distribution of snow covered area and how quickly the snow covered area drops with melt.

We agree that the "full snowpack" control on early-summer streamflow is not convincing, and we have amended the text to reflect this large unconstrained difference. However, we maintain that the other deeper snow metrics are useful for understanding what depths might persist at different points in the season. Visualizing Fig. 12 as a timeseries is interesting because it shows that the watershed become "most different" around late August (full snowpack panels), and that the 90% metric explains streamflow variations consistently across ~2 months.

More importantly, reserving interpretation of these graphs to places where all of the lines cross for different threshold values seems just to say that the deepest snow accumulations (>90 to 95 percentile) are a control on the differences between the 5 basins. Couldn't we just take the top left plot of Figure 11 and plot a mean August "runoff efficiency relative to initial snowpack" versus metrics of snow heterogeneity, like the volume of snow above the 95 percentile? That seems like it would be a more direct illustration of what you are trying to say. It also looks like you would get a similar answer but with less pronounced slope when considering the counterfactual (lower left). This is an approach, where seeing a great deal of variability in outcome, one can use covariates to explain that variation.

This suggestion has been implemented in the form of a new figure (Fig. 13), which shows correlations between August-September streamflow (normalized by total snow above 3000 m) and various other metrics. We exclude the counterfactual from this plot for greater simplicity as it is already addressed in Figs. 11-12.

This manuscript represents an impressive effort in data reduction to generate variables useful in a number of analyses that are threaded together. It is exemplary in some respects in terms of serving up multiple perspectives on the snow to flow problem. For the curious and persistent, there is much to see and it is relatively transparent. That being said, there is so much presented, some with much detail (maybe more than needed in some places), that it is a bit overwhelming to take in and be able to understand the effects of choices made during the analysis. I'm wondering if there are opportunities that could just use simpler bits of data from Tables 1 and 4, and Figure 9 for bivariate plots using the 5 basins to articulate the differences among the basins, and better pin down the contribution of snowpack variability and abundance of PSI.

We agree that it is hard to optimally organize such a large amount of material. We have made an effort to condense the key findings (i.e., the interrelationship hinted at in the title) into the new Fig. 13, with accompanying discussion text tying everything together.

Thank you for the opportunity to share some thoughts about my experience reading the paper with the authors. Hopefully they can draw on the reflections to help them better communicate what they wanted to communicate.

We thank the reviewer for their time and hope that our edits have enhanced the experience for future readers.